# UNCERTAINTYRAG: SPAN-LEVEL UNCERTAINTY ENHANCED LONG-CONTEXT MODELING FOR RETRIEVAL-AUGMENTED GENERATION

## ABSTRACT

We present *UncertaintyRAG*, a novel approach for long-context Retrieval-Augmented Generation (RAG) that utilizes Signal-to-Noise Ratio (SNR)-based span uncertainty to estimate similarity between text chunks. This span uncertainty enhances model calibration, improving robustness and mitigating semantic inconsistencies introduced by random chunking. Leveraging this insight, we propose an efficient unsupervised learning technique to train the retrieval model, alongside an effective data sampling and scaling strategy. *UncertaintyRAG* outperforms baselines by 2.03% on LLaMA-2-7B, achieving state-of-the-art results while using only 4% of the training data compared to other advanced open-source retrieval models under distribution shift settings. Our method demonstrates strong calibration through span uncertainty, leading to improved generalization and robustness in long-context RAG tasks. Additionally, *UncertaintyRAG* provides a lightweight retrieval model that can be integrated into any large language model with varying context window lengths, without the need for fine-tuning, showcasing the flexibility of our approach.

## 1 INTRODUCTION

Large Language Models (LLMs) have shown impressive capabilities across various natural language tasks, including long-context question-answering (QA), where the model processes a lengthy text and a question to generate a response (Caciularu et al., 2022). The QA paradigm encompasses a wide range of tasks, such as commonsense reasoning (Yang et al., 2018; Geva et al., 2021) and mathematical reasoning (Cobbe et al., 2021; Amini et al., 2019). However, due to the limitations in computational resources and the model's lack of ability to extrapolate context length, handling long-context settings remains a challenge for large language models, although continuous improvements are being made in this area.

Recent advances seek to overcome this limitation by designing linear attention mechanisms (Katharopoulos et al., 2020; Wang et al., 2020; Gu & Dao, 2023), which improve the memory and time efficiency for long sequences. Pruning the KV cache (Ge et al., 2023; Zhang et al., 2024; Agarwal et al., 2024) and quantizing the KV cache (Hooper et al., 2024; Liu et al., 2024) are other approaches to enhancing long context generation by compressing the model's KV-cache, providing a training-free, lightweight solution to reduce memory and computational overhead. However, these methods are generally difficult to achieve context length extrapolation. Moreover, the aforementioned approaches usually truncate the long-context to just fit the context window of LLMs, which not only prevents LLMs from actually seeing the entire input text but also faces an unacceptable memory overhead in resource-constrained environments. The efficient positional encoding strategy (Liu et al., 2023a; Su et al., 2024) is a solution that can achieve length extrapolation, but it often requires training the entire LLMs.

Another lightweight solution for handling long contexts is to utilize long-context Retrieval-Augmented Generation (RAG) in conjunction with long-context chunking (Xu et al., 2023b; Jiang et al., 2024b; Sarthi et al., 2024; Xu et al., 2023a), which avoids the need for LLMs to have length extrapolation capability. Long-context retrieval-augmented generation refers to a method in natural language processing where a model retrieves relevant information from large external sources to assist in

generating responses. It extends the traditional RAG by handling much longer input contexts, enabling effective processing of broader and more detailed information for tasks like question answering, summarization, and document understanding. Typically, RAG employs existing LLMs with limited context windows to retrieve relevant chunks. These chunks are obtained by retrieval models, either with or without semantic truncation (Sarthi et al., 2024; Li et al., 2024a; Jiang et al., 2024b; Xu et al., 2023b;a). Retrieval models (Izacard et al., 2021; Xiao et al., 2023b; Chen et al., 2024; Lin et al., 2023a) are commonly used for this purpose; however, they require a large amount of high-quality labeled data for training, which limits their scalability and adaptability. Modern RAG systems may rely on complex chunking methods (Sarthi et al., 2024) and require LLMs to have relatively long context windows (Jiang et al., 2024b; Xu et al., 2024). Furthermore, the lack of labeled data to determine if (query, chunk) pairs are related (Lewis et al., 2020a;b) poses significant limitations for training retrieval models in RAG systems. Recent research combines RAG with long-context LLMs to handle extended contexts and mitigate semantic incoherence in chunk processing (Xu et al., 2023b; Jiang et al., 2024b; Xu et al., 2024; Li et al., 2024c; Luo et al., 2024; Sarthi et al., 2024; Duarte et al., 2024). Some of the latest work also attempts to inject retrieval capabilities into LLMs through training, enabling them to process and generate long contexts directly (He et al., 2024; Cheng et al., 2024). However, these approaches often require either sophisticated chunking strategies, which are time-consuming during inference, or fine-tuning adapters for specific LLMs to manage chunk representation compression. Additionally, the complexity of these methods makes them vulnerable to distribution shifts. Some methods also necessitate training an LLM with an extended context window or new architecture, which is highly resource-intensive. In contrast, improving a lightweight retrieval model that can be seamlessly integrated into various LLMs without additional training would be a more efficient solution.

Unlike previous work, our focus is on utilizing calibrated uncertainty quantification to estimate similarity between chunks, thereby training robust retrieval models within RAG. Specifically, we introduce a novel uncertainty estimation technique based on the Signal-to-Noise Ratio (SNR), which stabilizes predictions and reduces biases from random chunk splitting. Our analysis shows that when two chunks are concatenated and fed into the model to estimate similarity, the uncertainty measured by SNR can better reflect their alignment in the semantic space, which we have confirmed in our experiments. Building on this finding, we develop an unsupervised learning technique to train chunk embeddings. This method is decoupled from the long-context modeling capabilities of LLMs and can be adapted to any fixed-length context window, enhancing retrieval robustness without requiring additional fine-tuning or retraining of the LLM itself. By leveraging the LLM's calibrated self-information, we effectively measure similarity between text chunks, construct accurate positive and negative samples, and train a robust retrieval model. This approach enhances generalization under distribution shifts in long-context retrieval-augmented generation tasks. Moreover, it seamlessly integrates with existing methods by relying solely on external retrieval models, thus avoiding LLM-specific performance dependencies. Specifically, our contributions are as follows:

1. We propose a novel SNR uncertainty measurement technique to achieve better calibration by addressing prediction errors arising from random chunk splitting, thereby improving performance in similarity estimation between chunks.

2. We propose an unsupervised learning approach and train a retrieval model that outperforms strong open-source embedding models in long-context RAG tasks under distribution-shift settings.

3. We design an efficient data sampling strategy to scale data, enhancing our retrieval model training and significantly boosting performance. Compared to models such as BGE-M3, our method improves LLaMA-2-7B by 2.03% after SNR calibration while using only 4% of their data size.

4. We provide an in-depth analysis of the retrieval model, demonstrating continuous improvements across two key metrics. We explain how uncertainty measurement enhances chunk representation modeling and why data sample scaling contributes to improved performance.

## 2 RELATED WORK

### 2.1 ATTENTION MECHANISMS IN LONG CONTEXTS

An effective approach to facilitating long context is to avoid the $O(n^2)$ computational complexity of the standard attention mechanism by designing linear attention mechanisms, sparse attention

mechanisms, or low-rank attention mechanisms. These works can be categorized into the following four types: *i)* Sparse Attention Mechanisms: Reduce the computational burden of attention by exploiting inherent patterns within the attention mechanism (Jiang et al., 2024a; Ribar et al., 2023; Chen et al., 2023), or alternatively, by pruning the KV cache (Liu et al., 2023b; Xiao et al., 2023a; Pang et al., 2024; Zhang et al., 2024). *ii)* The attention mechanism with linear complexity: This typically involves transforming models with $O(n^2)$ complexity into $O(n)$ or $O(n \log n)$ linear attention (Zheng et al., 2022; Kitaev et al., 2020; Qin et al., 2022; Katharopoulos et al., 2020), or efficient long-sequence recurrent neural networks (Gu & Dao, 2023; Dao & Gu, 2024; Peng et al., 2023a; Yang et al., 2023). *iii)* Memory-augmented attention mechanisms: This typically involves encoding long-context text using additional memory blocks (He et al., 2024; Bertsch et al., 2024; Wang et al., 2024). *iv)* Hardware-friendly attention mechanisms: FlashAttention (Dao et al., 2022; Dao, 2023; Shah et al., 2024) accelerates precise attention computations by optimizing reads and writes across different levels of GPU memory. FlashAttention is especially effective for processing longer sequences. Among previous works, memory-augmented LLMs are most relevant to this study, as they involve memory retrieval. However, they typically require modifying the original model architecture and retraining the LLM or fine-tuning some parameters, which can be an impractical overhead in certain settings.

## 2.2 POSITION ENCODING

Extending the context window from scratch is challenging (Liu et al., 2023a; Su et al., 2024) because it requires significant computational resources. As a result, efficient position encoding methods (Peng et al., 2023b; Li et al., 2023a; Chi et al., 2022; Press et al., 2021; Peng & Quesnelle, 2023) have gained attention for improving length extrapolation. These works attempt to expand the context window on pre-trained LLMs using a small amount of data.

However, considering that many LLMs are closed-source, end-to-end training of LLMs with retrievers is impractical. Additionally, fully fine-tuning LLMs on long-context data remains costly. Fully fine-tuning models such as LLaMA (Touvron et al., 2023), especially with sequences longer than 16,000 tokens, is prohibitively expensive due to the quadratic time and memory complexities associated with precise attention mechanisms. Therefore, extending the model's context window by continual training through position encoding remains unacceptable.

Recent RAG systems (Xu et al., 2023b; 2024) combining RAG with long context LLMs have utilized position encoding to fine-tune LLMs for context window extrapolation, either requiring LLMs to handle retrieved chunks as long-context inputs (Jiang et al., 2024b; Xu et al., 2024) or flexibly combining the strengths of both through a routing mechanism (Li et al., 2024c). These works consider combining the strengths of long context LLMs and RAG. They demonstrate that, when resources are abundant, long context LLMs consistently outperform RAG in terms of average performance. However, the significantly reduced cost of RAG remains a notable advantage. They also find that long-context window LLMs can effectively alleviate the context fragmentation issue in RAG. However, further extending the context window of LLMs still is challenging, as it requires a significant amount of computational resources. Therefore, how to expand the boundaries of RAG systems given the limited context window of LLMs remains a question worth exploring. Our work primarily focuses on optimizing the retrieval model in RAG, given the context window length of LLMs, to extend the boundaries of the RAG system's ability to handle long contexts.

## 2.3 RETRIEVAL-AUGMENTED GENERATION

The existing RAG frameworks heavily rely on the quality of the retrieval model. Due to retrieval model's context window limitations, they often tend to use short retrieval units in open-domain QA tasks (Lewis et al., 2020a; Karpukhin et al., 2020; Ni et al., 2021). These models rely on a bi-encoder and typically require manual query-passage annotations for training the encoder.

Moreover, some studies (Duarte et al., 2024; Luo et al., 2024; Sarthi et al., 2024; Li et al., 2023b) highlight the importance of appropriately chunking the input text for RAG systems. Complex chunking schemes, however, add inference complexity and increase latency. Therefore, a simpler chunking method is necessary to avoid this issue, ensuring robustness to different chunking. Existing methods (Jiang et al., 2024b; Xu et al., 2024) attempt to use LLMs capable of handling longer contexts to increase the length of retrieved chunks, thereby reducing the number of retrieval units and avoiding

the input of overly fragmented and incomplete information. This, in turn, alleviates the burden on the retrieval model in processing long contexts. Recently, a study (Luo et al., 2024) also proposes a three-stage fine-tuning approach to embed longer contexts into a special token, addressing the issue of semantic discontinuity caused by chunk splitting. However, their method requires complex training techniques for the retrieval model, making it difficult to easily scale the data size.

Distribution shift in RAG has also garnered attention (Li et al., 2024b; Sagirova & Burtsev, 2023). They use a calibrated model's confidence to detect "long-tailness" examples and implement an improved RAG pipeline. However, to address distribution shifts in the chunks of input, instance-level uncertainty estimation provides limited assistance to RAG, as it does not help LLMs retrieve long-tail knowledge. Additionally, their work focuses solely on improving the pipeline without enhancing the performance of the retrieval model itself.

In summary, the above work has the following issues: *i)* **Complex Chunking Overhead**: Complex chunking methods that may increase additional computational overhead during inference. *ii)* **High Cost of Labeled Data**: Requires labeled data between chunks and queries. This is usually costly in terms of extensive manual annotation effort for long-context retrieval tasks. *iii)* **Poor Handling of Semantic Incoherence**: Open-source embedding models often struggle to handle semantically incoherent chunks, making it difficult to generalize in long-context settings.

To address the aforementioned issues, we focus on facilitating representation learning in RAG under a simple long-context chunking setting. Considering the additional overhead introduced by complex chunking methods during the inference stage, we adopt a simple strategy of dividing chunks every 300 letters and develop an unsupervised learning technique to train a robust retrieval model capable of handling semantic discontinuities in this chunking approach under distribution shift scenarios. Additionally, we propose a method called span uncertainty measurement to construct training labels for the data and compare it with existing open-source embedding models (Izacard et al., 2021; Xiao et al., 2023b; Chen et al., 2024; Lin et al., 2023a) within the RAG system. Typically, these embedding models require extensive data for pre-training; however, we outperform them in distribution shift scenarios using far less training data than they require.

## 3 METHODOLOGY

In this section, we introduce a span uncertainty method based on SNR to obtain similarity scores between chunks. We then use these similarity scores to construct positive and negative samples for training our retrieval models. This process typically does not involve using query data for training, so we further develop two methods to scale chunk data to enhance the model's distribution shift generalization ability. Additionally, we provide insights into the effectiveness of these methods in the experimental section.

Figure 1: Each line in the figure represents the trend of SNR variation for different samples, where two chunks are concatenated and input into the LLM for uncertainty estimation. The SNR is calculated as a sliding window moves across the concatenated input. Notably, the SNR values exhibit a significant drop early on, even before reaching the end of the first chunk.

### 3.1 SPAN UNCERTAINTY

Recently, there has been widespread attention (Xiong et al., 2023; Ye et al., 2024) to the measurement of uncertainty in LLMs. Kuhn et al. (2023) leverage the natural language inference task to infer semantic entropy to calibrate the model's uncertainty. Gupta et al. (2024); Duan et al. (2024) adopt token-level uncertainty to achieve more fine-grained calibration. Lin et al. (2023b) measure the semantic equivalence of LLM responses based on model output probabilities, converting similarities into uncertainty measures. Inspired by these works, we input two chunks into LLMs, using the self-information of the model's output as a measure of uncertainty. To achieve better calibration, we draw inspiration from the sample gradient SNR statistic used to estimate generalization errors in Liu et al. (2020). We leverage this statistic to quantify uncertainty by calculating the SNR of the model's sample output probabilities. We first concatenate two chunks and input them into the model to obtain the probability. We then use this SNR to measure span uncertainty, and finally, convert the span uncertainty into a measure of similarity. Specifically, the uncertainty can be expressed by the

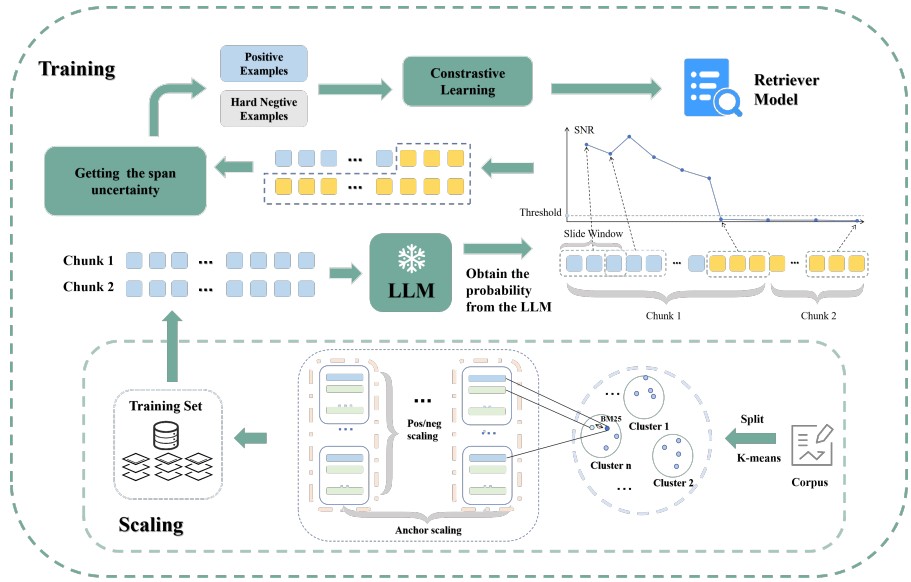

Figure 2: Scaling and Trainning. The figure presents the details of scaling and training.

self-information:

$$I(x_i|x_{i-1}, \ldots, x_0) = -\log p(x_i|x_{i-1}, \ldots, x_0), \tag{1}$$

where $x_i$ is the current token, and $p(x_i|x_{i-1}, \ldots, x_0)$ is the conditional probability of token $x_i$ given the preceding tokens $x_{i-1}, x_{i-2}, \ldots, x_0$ in the sequence. Previous effort (Duan et al., 2024) has suggested that not all tokens in auto-regressive LLM text equally represent the underlying meaning, as "linguistic redundancy" often allows a few keywords to convey the essence of long sentences. Additionally, we have observed a specific window in the model's output log probabilities. The probability distribution within this window is stable, which we believe indicates better calibration, a finding that is further validated in subsequent experiments. This stable behavior is intuitively characterized by the self-information of tokens stabilizing at a low value without significant fluctuations. As shown in Figure 1, we use a sliding window with a length of 20 to calculate the SNR within the window, with a sliding step of 10. We found that at a certain turning point, the SNR tends to stabilize. We choose this stable interval to measure the uncertainty of the model. **We define the span-level probabilities within this window as a measure of uncertainty and treat this confidence score as a special type of entailment relationship (Lin et al., 2023b), which represents a form of semantic equivalence.** Based on the above observations and the additive property of $I(x)$, we present the following definition:

$$\text{SNR}_j = \frac{\frac{1}{n_j} \sum_{i=1}^{n_j} I(x_{ij}|x_{(i-1)j}, x_{(i-2)j}, \ldots, x_{0j})}{\text{Var}\left(I(x_{ij}|x_{(i-1)j}, x_{(i-2)j}, \ldots, x_{0j})\right)}, \tag{2}$$

The span uncertainty, denoted as $SU(x)$, is defined as the average self-information over all concatenated tokens across the selected windows, where overlapping tokens are counted only once. The formula is:

$$SU(x) = \frac{\sum_{j=1}^m \mathbf{1}_{\text{SNR}_j < \sigma} \sum_{i=1}^{n_j} I(x_{ij}|x_{(i-1)j}, \ldots, x_{0j})}{\sum_{j=1}^m \mathbf{1}_{\text{SNR}_j < \sigma} n_j}, \tag{3}$$

where $\mathbf{1}_{\text{SNR}_j < \sigma}$ is an indicator function that equals 1 when the SNR of the $j$-th window is below the threshold $\sigma$, and 0 otherwise. $n_j$ represents the number of tokens in the $j$-th window, and $I(x_{ij}|x_{(i-1)j}, \ldots, x_{0j})$ denotes the self-information of token $x_{ij}$ given its preceding context. This helps in estimating the similarity between the two chunks of text by considering the uncertainty in the token sequences across those windows. Typically, the chosen token length varies with changes in the sliding window of the span.

## 3.2 Training Strategy

In this section, we focus on how we apply span uncertainty to construct positive and negative samples for all chunks in the training dataset. We do not consider incorporating the combinations of queries and chunks into the training strategy, yet the model still performs well. Then, we introduce a strategy for scaling chunk combinations, which involves scaling chunk anchors and scaling positive and negative samples after fixing chunk anchors.

### 3.2.1 Construction of Positive and Negative Samples

Given the limitations in inference computational efficiency, we avoid using overly complex segmentation strategies when constructing the training dataset. Instead, we uniformly set the chunk size to 300 letters for splitting the data. After shuffling the data, we combine each chunk with other chunks to construct a matrix $S$. Each element $S_{ij}$ in this matrix represents the span uncertainty estimated when $\text{ch}_i$ and $\text{ch}_j$ are combined sequentially and input into the LLM, indicating their degree of similirity. Due to the significant computational cost of estimating this matrix, we use BM25 (Robertson et al., 2009) as a score function and denote the BM25 score for the pair as $s_{\text{BM25}}(\text{ch}_i, \text{ch}_j)$. We then select the $M$ samples with the highest BM25 scores for each anchor $\text{ch}_i$ to estimate the $SU(x)$, resulting in the final sparse matrix $\hat{S}$. In formulaic terms, let $s_{\text{BM25}}(\text{ch}_i, \text{ch}_j)$ denote the BM25 score for the combination of chunks $i$ and $j$. The final sparse matrix $\hat{S}$ is given by:

$$\hat{S}_{ij} = \begin{cases} SU(\text{ch}_i, \text{ch}_j) & \text{if } s_{\text{BM25}}(\text{ch}_i, \text{ch}_j) \geq \text{Top}_M(s_{\text{BM25}}(\text{ch}_i, \text{ch}_M)), \\ 0 & \text{otherwise,} \end{cases} \quad (4)$$

where $\text{Top}_M(s_{\text{BM25}}(\text{ch}_i, \text{ch}_M))$ denotes the $M$-th highest BM25 score chunk $\text{ch}_M$ for anchor $\text{ch}_i$. We set up two windows within $M$, each containing $m$ samples. These samples are ranked by $\hat{S}_{ij}$ scores, with the top $m$ considered positive samples and the bottom $m$ considered negative samples. We sample positive and negative samples within these two windows, denoted as $\text{ch}_i^+$ and $\text{ch}_i^-$, respectively.

### 3.2.2 Data Scaling Strategy

All our experiments are conducted under distribution shift setting, meaning our retrieval model needs to handle semantically disjointed chunks resulting from segmented text, as well as noisy data arising from long-tail distributions that are never encountered in the trainning dataset. Our span uncertainty effectively calibrates the model's uncertainty measurement using the SNR metric, identifying meaningful output probabilities to construct an anchor-positive-negative sample dataset.

**Anchor Sample Scaling Strategy** We refer to each chunk as an anchor sample. We initially merge the five datasets: HotpotQA (Yang et al., 2018), MultiFieldQA (Bai et al., 2023), Qasper (Dasigi et al., 2021), NarrativeQA (Kočiský et al., 2018), and QMSum (Zhong et al., 2021). Each dataset is split based on spaces, and every 300 letters are grouped into a chunk. As a result, we obtain 37,799, 14,547, 15,865, 72,146, and 38,398 chunks, respectively. Due to computational limitations, it is challenging to score all possible combinations of such a large number of chunks. Therefore, we first use k-nearest neighbors (KNN) to cluster each dataset into $k$ clusters. Then, we randomly select $c$ chunks from each cluster, resulting in $k*c$ chunks per dataset. By mixing the five datasets, we obtain a total of $50*c$ chunks. By increasing the multiple of $c$, we scale the total number of anchor samples.

**Positive and Negative Sample Scaling Strategy** When a sample anchor $\text{ch}_i$ is selected, we need to concatenate it with other chunks and input them into the LLM for scoring. Due to the high computational cost, we employ a method similar to anchor sample scaling. First, using the KNN method, we randomly sample $n$ samples $\text{ch}_j$ from each of the $k$ clusters, where the clusters are formed from the previously mentioned $50*c$ chunks. This way, we obtain $10*n$ samples in a single sampling process. Then, we use BM25 to score the anchor and each sample, $s_{\text{BM25}}(\text{ch}_i, \text{ch}_j)$, and select the final $M$ samples. These are scored using LLMs to obtain the final $\text{ch}_i^+$ and $\text{ch}_i^-$. Finally, we obtain a triplet $(\text{ch}_i, \text{ch}_i^+, \text{ch}_i^-)$. By repeating this sampling process, we can scale the number of positive and negative samples for the anchor $\text{ch}_i$.

### 3.2.3 Contrastive Learning

We design a contrastive learning strategy to train our retrieval model:

$$\mathcal{L}_{2K-1} = -\log \frac{e^{f\left(\text{ch}_i, \text{ch}_i^+\right)}}{e^{f(\text{ch}_i, \text{ch}_i^+)} + \sum_{j=1}^{2K-2} e^{f(\text{ch}_i, \text{ch}_j)} + e^{f(\text{ch}_i, \text{ch}_i^-)}}, \tag{5}$$

where $\text{ch}_i$ represents the anchor chunk, which is the chunk for comparison. The term $\text{ch}_i^+$ represents a positive chunk, which is a chunk similar to $\text{ch}_i$. $\text{ch}_i^-$ denotes a hard negative chunk, which is a sequence that is dissimilar to $\text{ch}_i$. The remaining negative chunks are represented by $\text{ch}_j$, where $j$ ranges from 1 to $2K - 2$, which are the positive and negative chunks of other anchor chunks from the current batch. The similarity $f\left(\text{ch}_x, \text{ch}_y\right)$ between the anchor chunk $\text{ch}_x$ and another chunk $\text{ch}_y$ is calculated as the inner product of their chunk embeddings of BERT (Devlin, 2018):

$$f\left(\text{ch}_x, \text{ch}_y\right) = \langle b_e(\text{ch}_x), b_e(\text{ch}_y)\rangle, \tag{6}$$

where $b_e(\text{ch}_x)$ and $b_e(\text{ch}_y)$ represent the embeddings of the chunks $\text{ch}_x$ and $\text{ch}_y$, respectively.

### 3.3 MODEL INFERENCE

Due to our model's focus on using model uncertainty to promote the unsupervised learning aspect of retrieval models, after training and obtaining the embedding model, we can input long-contexts to the embedding model for retrieval after every 300-letters chunk, without re-rank part in our pipeline. Therefore, our inference stage is efficient, specifically divided into the following steps: *i*) We directly chunk the input document based on a fixed length, which is efficient, without using complex chunking mechanisms. *ii*) Input the query to the retrieval model to retrieve the most similar *m* chunks. *iii*) Input the retrieved *m* chunks and the query to the model to produce the final answer.

## 4 EXPERIMENT

| Engine | Model | Truncate | BERT | Contriever | BGE-Large | BGE-M3 | GRAGON-PLUS | All Chunking | Precise Chunking | Ours |
|---|---|---|---|---|---|---|---|---|---|---|
| | 2WikiMultihopQA | 28.50 | 32.73 | 33.60 | 34.14 | 29.64 | 34.61 | 33.77 | 34.60 | **37.27** |
| | Musique | 9.41 | 18.74 | 14.25 | 24.2 | **24.27** | 20.50 | 20.00 | 16.53 | 23.03 |
| LLaMA-2-7B-Chat-HF | TREC | 64.50 | 66.00 | 70.00 | 70.05 | **71.00** | 70.50 | 67.50 | 65.50 | 68.00 |
| | TriviaQA | 77.80 | 78.69 | 76.09 | 75.10 | 75.74 | 77.51 | 78.47 | 79.84 | **80.41** |
| | SAMSum | 40.45 | 40.01 | 38.45 | 40.34 | 40.37 | 41.08 | 40.72 | 41.32 | **42.49** |
| | Average | 44.13 | 47.23 | 46.48 | 48.85 | 48.20 | 48.84 | 48.09 | 47.56 | **50.23** |
| | 2WikiMultihopQA | 22.74 | 23.19 | 23.02 | 23.41 | 23.81 | 22.21 | 20.89 | 23.50 | **24.69** |
| | Musique | 7.55 | 12.64 | 10.12 | 14.84 | **15.06** | 13.01 | 13.34 | 12.50 | 13.62 |
| Vicuna-7B | TREC | 67.50 | 67.50 | **70.50** | **70.50** | 69.00 | 70.00 | 65.50 | 69.00 | 69.00 |
| | TriviaQA | 75.06 | 73.21 | 74.72 | 74.69 | 74.39 | 73.67 | 79.00 | 75.04 | **76.34** |
| | SAMSum | 37.14 | 36.54 | 35.81 | 36.04 | 36.83 | 37.22 | **38.73** | 37.03 | 37.88 |
| | Average | 42.20 | 42.61 | 42.83 | 43.89 | 43.82 | 43.22 | 43.29 | 43.41 | **44.31** |
| | 2WikiMultihopQA | 23.15 | 27.80 | 28.33 | 29.50 | 27.47 | 29.51 | 27.89 | 27.53 | **29.88** |
| | Musique | 8.11 | 13.12 | 9.88 | 14.15 | 13.02 | 13.24 | 13.10 | 12.78 | **14.94** |
| Vicuna-7B-16K | TREC | 66.50 | 67.50 | 69.50 | 69.00 | 69.00 | **70.00** | 67.00 | 66.00 | 68.50 |
| | TriviaQA | 83.96 | 82.93 | 85.74 | 84.56 | 84.46 | 83.05 | 84.84 | 84.71 | **84.98** |
| | SAMSum | 7.80 | 20.75 | 19.88 | 19.32 | 19.50 | 20.67 | **20.86** | 19.22 | 19.77 |
| | Average | 37.91 | 42.42 | 42.51 | 43.30 | 42.68 | 43.29 | 42.74 | 42.05 | **43.62** |
| | 2WikiMultihopQA | 34.73 | 35.82 | 40.28 | **42.05** | 38.13 | 38.46 | 36.15 | 37.31 | 38.28 |
| | Musique | 12.12 | 22.13 | 18.69 | 24.69 | **24.74** | 22.71 | 19.79 | 21.30 | **24.74** |
| LLaMA-2-13B-Chat-HF | TREC | 69.50 | 67.00 | 70.00 | 70.00 | **70.50** | 70.00 | 67.00 | 68.50 | 68.50 |
| | TriviaQA | 80.58 | 80.86 | 78.27 | 76.52 | 77.95 | 79.69 | 81.50 | 81.41 | **82.47** |
| | SAMSum | 36.95 | 41.62 | 41.00 | 42.49 | 42.43 | 41.23 | **43.09** | 41.03 | 42.61 |
| | Average | 46.77 | 49.48 | 49.65 | 51.14 | 50.75 | 50.37 | 49.50 | 49.91 | **51.32** |

Table 1: Experiment results on long-context retrieval augmented language generation. *All Chunking* denotes the use of the average self-information of all tokens from two concatenated chunks as the similarity score for selecting positive and negative samples in contrastive learning. In contrast, *Precise Chunking* denotes the accurate segmentation of the chunks, utilizing only the average self-information from the second chunk.

We primarily train our retrieval model on the HotpotQA, MultiFieldQA, Qasper, NarrativeQA, and QMSum datasets. We evaluate the model on 2WikiMultihopQA (Ho et al., 2020), Musique (Trivedi et al., 2022), TREC (Li & Roth, 2002), TriviaQA (Joshi et al., 2017), and SAMSum (Zhong et al., 2021). Under this distribution-shift setting, our retrieval model has never been exposed to any data from these test datasets. We provide a detailed introduction to our dataset in Appendix A. We provide detailed settings of the hyperparameters in Appendix B. We adopt various open-source embedding models as our baselines, which typically involve pre-training on large datasets. An introduction to the baselines is provided in Appendix C. In Appendix E, we demonstrate the advantages of our method in terms of an efficient training data sampling strategy and parameter efficiency. In our evaluation,

| Model | Scaling method | 2WikiMultihopQA | Musique | TREC | TriviaQA | SAMSum | Average |
|---|---|---|---|---|---|---|---|
| All Chunking | w/o Scaling | 23.03 | 23.91 | 66.00 | 79.15 | 41.77 | 46.86 |
| | Pos/neg Scaling | 32.47 | 22.41 | 66.00 | 79.15 | 41.86 | 48.38 |
| | Anchor Scaling | 34.87 | 21.49 | 67.50 | 77.46 | 40.72 | 48.41 |
| | Anchor and Pos/neg Scaling | 33.77 | 20.00 | 67.50 | 78.47 | 40.72 | 48.09 |
| Precise Chunking | w/o Scaling | 35.65 | 18.91 | 66.00 | 78.10 | 41.22 | 47.97 |
| | Pos/neg Scaling | 33.77 | 19.09 | 64.50 | 79.48 | 41.89 | 47.74 |
| | Anchor Scaling | 34.81 | 16.76 | 67.50 | 78.00 | 41.64 | 47.89 |
| | Anchor and Pos/neg Scaling | 34.60 | 16.53 | 65.50 | 79.83 | 41.32 | 47.55 |
| Uncertainty-RAG | w/o Scaling | 36.05 | 19.58 | 68.00 | 79.98 | 42.25 | 49.17 |
| | Pos/neg Scaling | 37.86 | 22.31 | 65.50 | 79.83 | 42.54 | 49.61 |
| | Anchor Scaling | 36.94 | 20.73 | 69.00 | 78.73 | 42.09 | 49.50 |
| | Anchor and Pos/neg scaling | 37.27 | 23.03 | 68.00 | 80.41 | 42.49 | 50.23 |

Table 2: Experiment results on scaling of anchor and positive/negative samples. In the table, the positive/negative samples are denoted as pos/neg.

we use four models: Llama2-7B/13B-chat-hf (Touvron et al., 2023), Vicuna-7B/7B-16K (Chiang et al., 2023).

## 4.1 MAIN RESULT

As shown in Table 1, the following observations can be made: *i*): Robust RAG retrieval model can significantly improve the performance of the 4K context window LLMs; however, this heavily depends on the performance of the retrieval model. *ii*): A powerful retrieval model can enhance the performance of LLMs. Vicuna-7B and LLaMA-2-7B-Chat-HF perform similarly within their 4K context windows, with an average performance difference of about 1.93% without retrieval augmentation. When equipped with a robust retrieval model, the average performance gap can increase to 6.14%. *iii*): Our method achieves the highest average performance, surpassing some open-source embedding models trained on large datasets. However, performance is limited on few-shot learning tasks like TREC due to the need for precise segmentation of in-context exemplars. Still, our method shows a consistent 2% improvement over the baseline. In addition, we have the following important findings.

**Analysis of The Impact of Span Uncertainty through SNR Calibration** In Table 1, our analysis reveals that the span uncertainty method, following SNR calibration, demonstrates significant improvements compared to both *All Chunking* and *Precise Chunking*. This enhancement is primarily due to the incorporation of self-information from certain tokens in the first chunk, which comes from the flexible sampling of spans across the two chunks by SNR. Thus, we can draw the following conclusions. *i*): SNR calibration effectively captures the similarity between the two chunks. *ii*): The tokens within the span sampled by SNR play a crucial role in assessing the relationship between the two chunks, suggesting that future research could benefit from focusing on span-level uncertainty.

**Ablation Study on The Effects of Scaling Up the Data** In Table 2, we present the results of scaling the anchor data in our dataset, fixing the anchor while scaling the positive/negative sample data, and scaling both the anchor and positive/negative samples simultaneously. Due to limitations in computational resources, we only doubled the size of the data. *i*): By simultaneously doubling both the anchor and positive/negative samples after fixing the anchor, we achieved state-of-the-art results, demonstrating the scalability of our method with respect to data size. *ii*): Our dataset sampling method is simple and easy to use, and applicable to any text dataset that can be chunked.

**Ablation Study on the Length and Number of Retrieved Chunks** In Table 3, we evaluate the Vicuna-7B-16K model on the 2WikiMultihopQA dataset, varying the number of chunks while keeping chunk length constant, and adjusting the chunk length while keeping the number of chunks constant. *i*): We find that increasing the number of chunks is not effective; instead, increasing the chunk length itself yields better results. *ii*): However, simply increasing the chunk size is not a one-size-fits-all solution, as it can also harm performance, which is contrary to the findings of LongRAG (Jiang et al., 2024b).

## 4.2 REPRESENTATION ANALYSIS

**Representation Similarity Analysis** In this part, we demonstrate the changes in similarity across different layers of our representation in Figure 3. Representation Similarity Analysis (RSA;Laakso & Cottrell (2000)) is a technique for measuring the similarity between two different representation spaces for a given set of stimuli. We demonstrated on 2WikiMultihopQA and the typical in-context

| Model | Chunk size | Chunk number | Length | Contriever | BGE-Large | BGE-M3 | DRAGON-PLUS | All Chunking | Precise Chunking | Ours |
|---|---|---|---|---|---|---|---|---|---|---|
| Vicuna-7B | 300 | 40 | 4k | 22.82 | 24.13 | 23.45 | 21.15 | 19.87 | 23.42 | 24.16 |
| | 300 | 30 | 3k | 23.02 | 23.41 | 23.81 | 22.21 | 20.89 | 23.50 | 24.69 |
| | 300 | 20 | 2k | 21.56 | 23.96 | 26.24 | 22.00 | 20.13 | 23.65 | 23.83 |
| Vicuna-7B-16K | 300 | 40 | 4k | 27.03 | 27.98 | 26.60 | 28.45 | 24.73 | 24.53 | 25.83 |
| | 300 | 30 | 3k | 27.26 | 28.93 | 27.98 | 29.45 | 25.81 | 24.88 | 26.56 |
| | 300 | 20 | 2k | 28.33 | 29.50 | 27.47 | 29.51 | 27.89 | 27.53 | 29.88 |
| | 600 | 40 | 8k | 26.94 | 28.83 | 26.78 | 27.11 | 26.70 | 27.57 | 25.83 |
| | 600 | 30 | 6k | 27.18 | 28.86 | 27.84 | 28.53 | 26.66 | 26.78 | 28.34 |
| | 600 | 20 | 4k | 27.72 | 28.93 | 29.08 | 29.26 | 27.78 | 27.75 | 29.64 |

Table 3: Abalation of different chunk size and different chunk number on 2WikiMultihopQA. Chunk number refers to the number $m$ of the most similar chunks selected as prompts. Length refers to the total length of the prompt.

learning task GSM8K (Cobbe et al., 2021) the changes in representations across different layers compared to the original BERT representations.

We observe that: *i:*) Before the eighth layer, variability in model representations is minimal across datasets and scoring methods. *ii:*) After the eighth layer, the model shows smaller changes on the GSM8K dataset with significant distribution shift, larger changes on the 2WikiMultihopQA dataset with minor shifts, and the most substantial changes on the training set. This indicates that RSA is an effective metric for assessing distribution shifts. *iii:*) Under the enhanced long-context modeling with Span Uncertainty, the model exhibits greater representation changes across varying distribution shifts, consistently outperforming precise chunking and leading

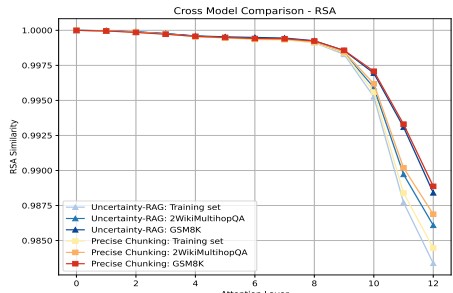

Figure 3: Representation Similarity Analysis

to performance improvements. To further analyze these representation variations, we will employ two metrics in the next part.

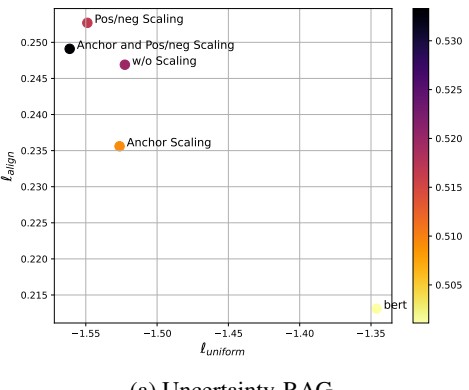

(a) Uncertainty-RAG

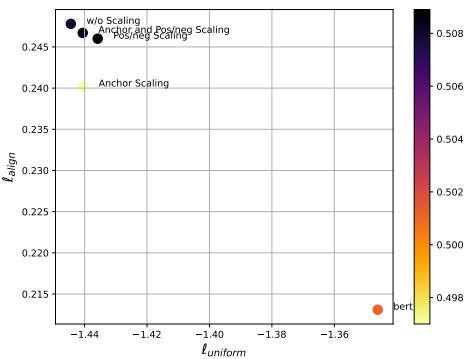

(b) Precise Chunking

Figure 4: Align and Uniform. This figure shows uniformity and alignment of different chunk embedding along with their averaged semantic textual similarity (STS (Conneau et al., 2017)) results.

**Representation Property Analysis** Wang & Isola (2020) introduces the properties of uniformity and alignment that contrastive learning improves in retrieval models, which are essential for evaluating the quality of retrieval models in machine learning. Alignment is quantified by the expected distance between the embeddings of paired positive chunk, assuming that these embeddings are normalized. The formula for alignment is expressed as:

$$\ell_{align} \triangleq \mathbb{E}_{(ch,ch^+)\sim p_{pos}} \left\| f(ch) - f(ch^+) \right\|^2, \tag{7}$$

On the other hand, uniformity assesses how evenly the embeddings are distributed across the embedding space. It is calculated using the expression:

$$\ell_{uniform} \triangleq \log \mathbb{E}_{ch_x,ch_y \overset{\text{i.i.d.}}{\sim} p_{data}} e^{-2\|f(ch_x)-f(ch_y)\|^2}, \tag{8}$$

where $p_{pos}$ represents positive pairs (similar chunks), and $p_{data}$ refers to the distribution of data

points from the dataset, with samples drawn independently. These measures align with the goals of contrastive learning, which aims to bring embeddings of similar chunks closer together while ensuring that embeddings of unrelated chunks $ch_x$ and $ch_y$ remain well-separated in the embedding space. Figure 4, shows the uniformity and alignment of different sentence embedding models along with their averaged STS results. We have the following findings: *i*): When we scale the positive and negative samples, the uniformity of the retrieval model increases, which aligns with the idea of InfoNCE (Oord et al., 2018), where increasing the noise samples can improve the uniformity of representations. *ii*): When we scale the anchors, the alignment of retrieval improves, which means we can align the retrieval model by performing data selection on the anchor samples of the model. *iii*): Enhancing the alignment level of the model does not improve the performance of the retrieval model, while improving the model's uniformity can enhance the performance of the retrieval model.

| Method | 2WikiMultihopQA | Musique | TREC | TriviaQA | SAMSum | Average |
|---|---|---|---|---|---|---|
| Minimum | 28.04 | 15.25 | 66.50 | 77.21 | 40.35 | 45.47 |
| Average | 31.55 | 17.72 | 66.50 | 78.21 | 41.56 | 47.10 |
| Log-sum | 29.77 | 16.25 | 69.50 | 80.21 | 41.55 | 47.25 |
| Entropy | 31.95 | 16.17 | 69.00 | 79.70 | 40.99 | 47.56 |
| Self-information | 34.60 | 16.53 | 65.50 | 79.84 | 41.32 | 47.56 |
| Ours | 37.27 | 23.03 | 68.00 | 80.41 | 42.49 | 50.23 |

Table 4: Comparison results of different uncertainty modeling approaches.

### 4.3 UNCERTAINTY MODELING

**Comparison of Uncertainty Modeling Approaches** In this part, we present the results of different uncertainty measurement approaches. We present different uncertainty measurement approaches in Appendix G, and the final experimental results in Table 4. We find that the performance of our method far exceeds simple token-level uncertainty estimation and sentence-level uncertainty estimation.

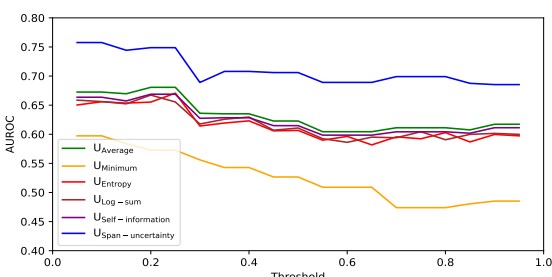

Figure 5: AUROCs of Uncertainty Measures. The horizontal axis represents the threshold $\tau$.

**Uncertainty Calibration Evaluation**
In this part, we use the AUROC (Lin et al., 2023b; Huang et al., 2024) metric as the evaluation standard for the uncertainty calibration of LLMs. Referring to their method, we introduce an ad hoc threshold $\tau \in \mathbb{R}$ to map the real-valued output of the deterministic correctness function to binary labels, i.e., $F_\tau(x, \hat{y}) = \mathbf{1}[F_1(x, \hat{y}) \geq \tau]$, where $x$ represents the model's input chunks and $\hat{y}$ represents the model's sampled output. $F_1$ represents the accuracy assessment of the model's output $\hat{y}$. Specifically, if the correctness function $F_1$ exceeds the threshold $\tau$, it is considered correct; otherwise, it is considered incorrect. $F_\tau$ is regarded as the final binary classification result where the model is confident. It is used to compare with $SU(\text{ch}_1, ..., \text{ch}_i, \text{query}, \hat{y})$ to obtain the final AUROC. The final results are presented in Fig 5, where we can see that the calibration of our uncertainty measures is significantly better than other methods. Both results are evaluated on the Llama-2-7b-chat-hf model using the TriviaQA (Joshi et al., 2017) dataset. The detailed introduction of the pipeline can be found in Appendix G.

## 5 CONCLUSION

In conclusion, *UncertaintyRAG* introduces a novel SNR-based span uncertainty approach to improve calibration in long-context Retrieval-Augmented Generation (RAG). The method outperforms powerful open-source embedding models like BGE-M3 while using only 4% of the training data. By leveraging span uncertainty, UncertaintyRAG promotes unsupervised learning, reducing the need for large labeled datasets, making it scalable and efficient. The results show that even with minimal data, UncertaintyRAG achieves superior performance and generalization under distribution shift scenarios. This lightweight solution requires no fine-tuning and integrates seamlessly into any LLM, making it versatile for various long-context tasks. Looking forward, *UncertaintyRAG* paves the way for optimizing retrieval models through advanced uncertainty quantification, particularly in resource-constrained environments.

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

APPENDIX

# A   TRAINING DATASET

In this work, we construct a new dataset from five existing datasets that include single-document QA, multi-document QA, and summarization tasks to train our dense retriever. All datasets are sourced from LongBench(Bai et al., 2023).

**HotpotQA**(Yang et al., 2018) is a challenging benchmark for multi-hop question answering, requiring models to retrieve and integrate information from multiple documents to arrive at the correct answer. It is specifically designed to evaluate a model's capacity for compositional reasoning, as questions often necessitate combining evidence from disparate sources. Additionally, HotpotQA includes supporting fact annotations, which serve to explain the reasoning process, making it suitable for tasks involving both factual retrieval and logical inference in natural language understanding.

**MultiFieldQA-en**(Bai et al., 2023) is manually curated to better evaluate the model's ability to understand long contexts across various fields. The documents contain evidence from multiple sources, such as legal documents, government reports, encyclopedias, and academic papers, which are randomly distributed to prevent biases that might arise from their placement at the beginning or end of the documents.

**Qasper**(Dasigi et al., 2021) is a question-answering dataset specifically tailored for academic papers in the field of natural language processing. It consists of questions generated by annotators who read full research papers and pose queries that can be answered using the information within the paper. The dataset is designed to evaluate a model's ability to perform non-factoid QA on scientific texts, involving tasks such as understanding methodologies, interpreting results, and synthesizing information from complex, domain-specific content.

**NarrativeQA**(Kočiskỳ et al., 2018) is a well-known question-answering dataset that includes full books from Project Gutenberg and movie scripts sourced from various websites. In this task, the provided passage is transcribed from books and often contains noise. The model's job is to generate a concise phrase by reasoning through the lengthy and noisy text.

**QMSum**(Zhong et al., 2021) is a query-based summarization dataset that includes transcripts of meetings and their corresponding summaries across various domains, including academia and industrial products. In this task, a meeting dialogue transcript is provided along with a question that prompts summarization of a specific topic within the dialogue. The answers typically consist of a few sentences.

And then we test on five dataset:

**2WikiMultihopQA**(Ho et al., 2020) is a benchmark for multi-hop question answering, which requires models to perform reasoning over multiple pieces of information to answer a single question. It contains questions that are designed to require information from multiple Wikipedia articles to answer correctly, promoting a more complex and connected reasoning process.

**Musique**(Trivedi et al., 2022) is a dataset designed for multi-hop question answering. Unlike HotpotQA, Musique demands more integrated reasoning by reducing possible shortcuts in reasoning, minimizing overlap between training and test data, and featuring more challenging distractor contexts. As a result, Musique is a significantly more difficult task than HotpotQA and is much less susceptible to shortcuts.

**TREC**(Li & Roth, 2002) is a widely used dataset for evaluating information retrieval and semantic search systems, encompassing documents from multiple domains. The answers are assessed for their relevance and accuracy based on predefined criteria, providing a robust framework for evaluating the performance of retrieval and question-answering models.

**TriviaQA**(Joshi et al., 2017) is a widely used question-answering dataset that provides complex, real-world questions paired with relevant documents from sources like Wikipedia and web pages. It is designed to test a model's ability to locate and understand the information needed to answer the questions, making it suitable for tasks such as machine reading comprehension and open-domain question answering.

**SAMSum**(Zhong et al., 2021) is a specialized dataset designed for the task of abstractive dialogue summarization. Developed to support research in natural language processing, particularly in generating concise summaries from conversational text, the SAMSum dataset contains dialogues that mimic everyday conversations one might find in messaging apps or casual settings.

## B  HYPERPARAMETER SETTINGS

In our experiments, we employed Llama-2-7b-chat-hf to compute uncertainty scores across different chunks. To identify the stable region of the Signal-to-Noise Ratio (SNR), we applied a sliding window technique, setting the window size to 20 and shifting by 10 steps at each interval. We consider the SNR to have reached a stable region when its value within a window falls below a predefined threshold, typically set at 2 or 3.

During the data scaling process, we set the number of clusters $k$ to 10, set $c$ to 800 or 1600 samples from each category and set sample number $n$ to 100.

During retriever training, we utilize the Adam optimizer (Kingma, 2014) with a batch size of 16, a learning rate of 1e-5, linear scheduling with warm-up, and a dropout rate of 0.1. And we run training on 8 NVIDIA A800 GPUs.

## C  BASELINES

There has been extensive research on Retrieval-Augmented Generation (RAG), with one critical aspect being the development of a robust representation model that effectively clusters sentence information with query information in high-dimensional space. In our work, we compare our approach with several influential and open-source methods from prior studies. This section provides a detailed overview of these methods.

**Contriever**(Izacard et al., 2021) is a dense retrieval model designed to improve performance and generalization through unsupervised training on diverse datasets. It uses contrastive learning to generate embeddings, bringing semantically related query-document pairs closer together while pushing irrelevant pairs apart. This method allows Contriever to effectively handle tasks like semantic search and question answering, even in zero-shot scenarios. By relying on unsupervised learning, it adapts well to various domains without needing annotated data, making it versatile for information retrieval tasks.

**BGE**(Xiao et al., 2023b) is a large-scale multilingual text embedding model developed by the Beijing Academy of Artificial Intelligence (BAAI), designed for tasks such as semantic retrieval and ranking across multiple languages. Pre-trained on extensive multilingual datasets, BGE encodes text into dense vector representations optimized for semantic similarity tasks, where the relationship between query and document pairs is captured by comparing their vector embeddings.

The model is primarily trained using contrastive learning, which teaches it to minimize the distance between embeddings of relevant query-document pairs while maximizing the distance for irrelevant pairs. BGE embeddings are applied in dense retrieval, which focuses on understanding the semantic meaning of text rather than relying solely on keyword matching. This makes BGE especially effective for tasks that demand high precision in matching semantically similar content, such as cross-lingual or multilingual retrieval scenarios.

**BGE-M3**(Chen et al., 2024) is an advanced extension of the BGE model, designed to support multi-modal retrieval by integrating dense, lexical, and multi-vector retrieval methods. This hybrid approach allows BGE-M3 to tackle a broader range of retrieval tasks by combining precise keyword matching with deeper semantic analysis. The training process of BGE-M3 employs self-distillation, where intermediate layers are optimized to rank query-document pairs, enhancing both efficiency and flexibility. By leveraging a combination of sparse and dense embeddings, BGE-M3 is highly effective for large-scale, multilingual retrieval tasks.

**Dragon**(Lin et al., 2023a) is a dense retrieval model designed to enhance performance and generalization in dense retrieval tasks by incorporating diverse data augmentation techniques. This model is notable for its robust handling of both supervised and zero-shot environments, utilizing innovative

methods to manage various query types and relevance labels. By introducing multiple forms of data augmentation—specifically for queries and labels—it effectively generalizes to unseen data.

## D PROMPT TEMPLATES

| **2WikiMultihopQA and Musique** |
| --- |
| Answer the question based on the given passages. Only give me the answer and do not output any other words. The following are given passages. |
| {**Prompt**} |
| Answer the question based on the given passages. Only give me the answer and do not output any other words. Question: {**Question**} Answer: |
| **TREC** |
| Please determine the type of the question below. Here are some examples of questions. {**Prompt**} {**Question**} |
| **SAMSum** |
| Summarize the dialogue into a few short sentences. The following are some examples. {**Prompt**} {**Question**} |
| **TriviaQA** |
| Answer the question based on the given passage. Only give me the answer and do not output any other words. The following are some examples. {**Prompt**} {**Question**} |

Table 5: Prompt template.

The prompt templates employed are similar to those proposed by Bai et al. (2023), and are listed in Table 5.

## E ANALYSIS OF TRAINING DATA UTILIZATION AND PARAMETER EFFICIENCY

| Model | Label | Number | Parameter Size |
| --- | --- | --- | --- |
| Contriever | Unlabeled | - | 768M |
| BGE-Large | Unlabeled Labeled | 100 millions 8 millions | 326M |
| BGE-M3 | Unlabeled | 184 millions | 560M |
| Dragon | Unlabeled | 28 millions | 110M |
| Ours | Unlabeled | 7 millions | 110M |

Table 6: The comparison of the number of paired texts used for training.

As shown in Table 6, our method demonstrates significant advantages over baseline models in both the required training data and parameter size:

**Reduced Training Data:** While baseline models such as BGE-Large and BGE-M3 require hundreds of millions of paired texts (100 million and 184 million, respectively), our method only requires **7**

**million** paired texts. This efficiency allows for effective training even in scenarios with limited data availability.

**Compact Parameter Size:** Our model maintains a parameter size of **110 million**, comparable to Dragon, which also has the same parameter count but requires **28 million** paired texts for training. In contrast, other baseline models like Contriever and BGE-Large have significantly larger parameter sizes (768 million and 326 million, respectively), which may lead to increased computational resource requirements and longer training times.

In summary, our method demonstrates superior efficiency in training data utilization and parameter scaling compared to baseline models, facilitating a broader application in scenarios with limited labeled data.

## F   FORMULATIONS FOR DIFFERENT APPROACHES TO MODELING UNCERTAINTY

| Method | Formula |
|---|---|
| Minimum | $u = -\log(\min(z_1, z_2, \ldots, z_n))$ |
| Average | $u = -\log(\mathrm{Avg}(z_1, z_2, \ldots, z_n))$ |
| Log-sum | $u = -\sum_{i=1}^{n} \log(z_i)$ |
| Entropy | $u = -\sum_{i=1}^{n} z_i \cdot \log(z_i)$ |
| Self-information | $u = -\log\left(\sum_{i=1}^{n} \exp(z_i - z_k)\right)$ |

Table 7: Six methods of calculating the uncertainty $u$ of a free form output of length $n$. $z_i$ is a conditional probability with respect to $p(x_i \mid x_{i-1}, \ldots, x_0)$. In the table, $i$ represents the current token at the $i-th$ position in the vocabulary.

In this section, we present several different methods for uncertainty modeling, with specific formulations shown in Table 7. Each method provides unique insights into quantifying uncertainty, which is crucial for improving model robustness and decision-making in complex scenarios.

## G   PIPELINE OF UNCERTAINTY CALIBRATION EVALUATION

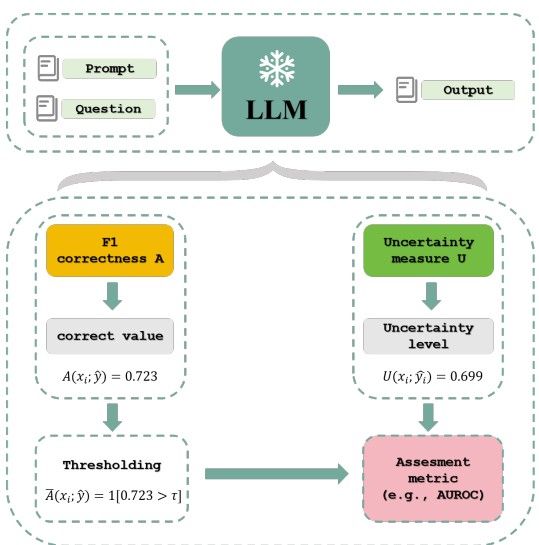

Figure 6: Common pipeline for assessing the quality of an LLM uncertainty measure.

In this section, we provide a detailed description of the process for calculating the AUROC, as illustrated in Fig 6. Similarly to previous work(Huang et al., 2024), the experiments are carried out on the TriviaQA dataset, using the Llama-2-7b-chat-hf model. First, we construct prompts suitable for TriviaQA in the format shown in Table 5 and input them into the LLM. At this stage, we obtain an output y from the model, as well as the logits corresponding to both the prompt and the output. For the output $\hat{y}$, we compute its score using a correctness function (in this experiment, we employ the F1-score). If this score exceeds a predefined threshold $F_\tau$, the input is considered correct; otherwise, it is deemed incorrect. Simultaneously, we evaluate the output logits using the uncertainty methods $U$ outlined in Table 7 to obtain an uncertainty score $U(x;\hat{y})$. Finally, we calculate the AUROC metric based on $F_\tau(x,\hat{y})$ and $U(x;\hat{y})$ to derive the final result.

## H  EXPERIMENTS ON DATASET SENSITIVITY

To validate the robustness of our method to the choice of data sets, we carried out another set of experiments with newly selected training and testing data sets. In the new experiments, the training set includes MultiFieldQA, Musique, QMSum, TREC, and TriviaQA, while the test set consists of NarrativeQA, Qasper, SAMSum, and 2WikiMultihopQA. The detailed experimental results are presented in Table 8

| LLaMA-2-7B-Chat-HF | Truncate | BERT | Contriever | BGE-M3 | BGE-Large | GRAGON-PLUS | Ours |
|---|---|---|---|---|---|---|---|
| 2WikiMultihopQA | 28.50 | 32.73 | 33.60 | 29.64 | 34.14 | 34.61 | **37.31** |
| NarrativeQA | 17.31 | 17.58 | 16.91 | 19.45 | 19.46 | 20.01 | **20.20** |
| Qasper | 18.14 | 19.07 | **21.01** | 20.31 | 19.93 | 20.50 | 20.81 |
| SAMSum | 40.45 | 40.01 | 38.45 | 40.37 | 40.34 | 41.08 | **41.26** |
| Average | 26.10 | 27.34 | 27.49 | 27.44 | 28.47 | 29.05 | **29.90** |

Table 8: Results with the new datasets.

All experiments were conducted on LLaMA-2-7B-Chat-HF. As observed, our method continues to achieve the best performance on the new datasets. This demonstrates the robustness of our method as it does not rely on the specific selection of data sets.

## I  EXPERIMENTS ON THE SENSITIVITY OF UNCERTAINTY MEASUREMENT MODELS

To validate the robustness of the model used to measure uncertainty, we conduct RAG experiments with the model using a retrieval model trained on uncertainty measurements obtained from Vicuna-7B with Vicuna-7B and LLaMA-2-7B-Chat-HF. The results are shown in Table 9 and Table 10.

| Dataset | Truncate | BERT | Contriever | BGE-M3 | BGE-Large | GRAGON-PLUS | Ours (Vicuna-7B) |
|---|---|---|---|---|---|---|---|
| 2WikiMultihopQA | 22.74 | 23.19 | 23.02 | 23.81 | 23.41 | 22.21 | **24.29** |
| Musique | 7.55 | 12.64 | 10.12 | **15.06** | 14.84 | 13.01 | 14.51 |
| TREC | 67.50 | 67.50 | **70.50** | 69.00 | **70.50** | 70.00 | 68.50 |
| TriviaQA | 73.21 | 73.21 | 74.72 | 74.39 | 74.69 | 73.67 | **78.49** |
| SAMSum | 36.54 | 36.54 | 35.81 | 36.83 | 36.04 | 37.22 | **39.64** |
| Average | 42.20 | 42.61 | 42.83 | 43.82 | 43.89 | 43.22 | **45.08** |

Table 9: The results of using Vicuna-7B as the uncertainty measurement model and testing on Vicuna-7B.

UncertaintyRAG demonstrates strong effectiveness in measuring uncertainty when combined with various models. Notably, using different models (including Vicuna-7B) to estimate uncertainty consistently improved RAG performance across experiments conducted on diverse models, such as Vicuna-7B and LLaMA-2-7B-Chat-HF. These results highlight the robustness and agreement of uncertainty measurement across different model architectures.

| Dataset | Truncate | BERT | Contriever | BGE-M3 | BGE-Large | GRAGON-PLUS | Ours (Vicuna-7B) |
|---|---|---|---|---|---|---|---|
| 2WikiMultihopQA | 28.50 | 32.73 | 33.60 | 29.64 | 34.14 | 34.61 | **35.74** |
| Musique | 9.41 | 18.74 | 14.25 | **24.27** | 24.20 | 20.50 | 22.68 |
| TREC | 64.50 | 66.00 | 70.00 | **71.00** | 70.50 | 70.50 | 67.50 |
| TriviaQA | 77.80 | 78.69 | 76.09 | 75.74 | 75.10 | 77.51 | **79.46** |
| SAMSum | 40.45 | 40.01 | 38.45 | 40.37 | 40.34 | 41.08 | **42.70** |
| Average | 44.13 | 47.23 | 46.48 | 48.20 | 48.20 | 48.84 | **49.62** |

Table 10: The results of using Vicuna-7B as the uncertainty measurement model and testing on LLaMA-2-7B-Chat-HF.

## J EXPLORATION OF USING SPAN UNCERTAINTY AS A RERANKING METHOD

To further explore the potential of our method in reranking, we first use BERT for coarse ranking to select candidate chunks. These chunks are then concatenated with the question and processed by an LLM. Our method is applied to measure similarity scores, which are used to rerank the chunks. Detailed results are shown in the Table 11.

| Dataset | BERT | After Re-ranking |
|---|---|---|
| 2WikiMultihopQA | 32.73 | 33.73 |
| Musique | 18.74 | 20.34 |
| TREC | 66.00 | 66.50 |
| TriviaQA | 78.69 | 79.78 |
| SAMSum | 40.01 | 41.24 |
| Average | 47.23 | 48.32 |

Table 11: Comparison of performance before and after reranking on LLaMA-2-7B-Chat-HF.

We can find that after re-ranking, the final results show a significant improvement. This could potentially be an area for further exploration in the future.

## K A SIMPLE CASE STUDY

Now, assume that we have two chunks:

- **Chunk A:** *How do you do?*
- **Chunk B:** *Thank you for taking the time to review our manuscript.*

Here, for the sake of convenience, we treat each word or punctuation mark as a token.

In this case, Chunk A consists of 5 tokens, and we can obtain the probabilities for these 5 tokens. Similarly, Chunk B consists of 11 tokens, for which we also have their probabilities. By concatenating these probabilities, we get:

$$P_{A1}, P_{A2}, \ldots, P_{A5}, P_{B1}, P_{B2}, \ldots, P_{B11}.$$

Assume the sliding window size is set to 3, with a stride of 1 (used here for clearer demonstration). We first calculate the SNR within the window:

$$(P_{B9}, P_{B10}, P_{B11})$$

and find that it is less than the set threshold.

We then slide the window one step to the left and calculate:

$$(P_{B8}, P_{B9}, P_{B10}),$$

and continue this process iteratively.

For this example, let us assume we eventually find that the SNR for:

$$(P_{A4}, P_{A5}, P_{B1})$$

exceeds the threshold. At this point, the sliding window stops.

We then take the probabilities of all tokens from the previous sliding windows (i.e., $P_{A5}$ to $P_{B11}$) and compute their average. This average value is used as a similarity score between the two chunks, forming positive and negative samples for subsequent contrastive learning training.

