# OpenReview forum: "UncertaintyRAG: Span Uncertainty Enhanced Long-Context Modeling for Retrieval-Augmented Generation"
_ICLR.cc/2025/Conference — Submitted to ICLR 2025_

### Official Review · Reviewer_jHwd · 2024-10-27

**Soundness:** 3
**Presentation:** 2
**Contribution:** 3
**Rating:** 5
**Confidence:** 4

**Summary:**

This work focuses on improving the dense retrieval model to facilitate retrieval-augmented generation (RAG).
This work introduces UncertaintyRAG, an innovative method that incorporates span uncertainty, grounded in Signal-to-Noise Ratio (SNR), to enhance the similarity estimation between text chunks.
The key strength of UncertaintyRAG is the robustness to generalize across different contexts.
This is achieved by the improved calibration mechanism in the measurement of span uncertainty.
Experiments with distribution shifts further demonstrate the robustness and the better performance of UncertaintyRAG.

**Strengths:**

1. This work introduces a novel method to improve the dense retrieval model. This method uses SNR-based span uncertainty to estimate similarity between text chunks.
2. Experimental results across various inference models and RAG tasks demonstrate that UncertaintyRAG outperforms baseline methods.
3. Further analysis reveals the inner properties of the trained dense retrieval model.

**Weaknesses:**

1. For measuring the span uncertainty, this work only considered one model (i.e., Llama-2-7b-chat-hf). As the proposed method is theoretically model-free, there should be experiments to demonstrate that (1) UncertaintyRAG performs well with various models to measure the uncertainty, or (2) various models can achieve an agreement on the uncertainty measurement.
2. The evaluation tasks should include single-doc tasks such as NQ and Qasper. To still keep the distribution shifts, you can move these two tasks from training set to the evaluation set, and add the three few-shot learning tasks into the training data.
3. There should be some comparisons with more baseline RAG methods, such as Self-RAG, FLARE, and RankRAG.

**Questions:**

I hope to see more experimental results to further demonstrate the effectiveness of UncertaintyRAG.
See the Weaknesses.

---

> ### Author Response · Authors · 2024-11-24
> **Response to Reviewer jHwd (Part 1/2)**
>
> Dear Reviewer jHwd,
>
> Thank you very much for your comment, we will address your concerns below.
>
> **Question 1:**
> For measuring the span uncertainty, this work only considered one model (i.e., Llama-2-7b-chat-hf). As the proposed method is theoretically model-free, there should be experiments to demonstrate that (1) UncertaintyRAG performs well with various models to measure the uncertainty, or (2) various models can achieve an agreement on the uncertainty measurement.
>
> **Response 1:**
> Thank you for your valuable suggestions regarding our experiments. Following your advice, we conducted additional experiments using Vicuna-7B to measure uncertainty and performed RAG experiments on Vicuna-7B. The results are presented in the table below.
>
>
> | Vicuna-7B    | Truncate |BERT	 | Contriever| BGE-M3| BGE-Large| GRAGON-PLUS|Ours(Vicuna-7B)|
> |  ----  | :----:  | :----:  | :----:  | :----:  | :----:  | :----:  | :----:  |
> | 2WikiMultihopQA|22.74		|23.19   |23.02      |23.81  |23.41     |22.21       |**24.29**
> | Musique    	 | 7.55		|12.64   |10.12      |**15.06**  |14.84     |13.01       |14.51
> | TREC			 |67.50		|67.50   |**70.50**      |69.00  |**70.50**     |70.00       |68.50
> | Triviaqa		 |73.21		|73.21   |74.72      |74.39  |74.69     |73.67       |**78.49**
> | SAMSum		 |36.54		|36.54   |35.81      |36.83  |36.04     |37.22       |**39.64**
> | Average		 |42.20		|42.61   |42.83      |43.82  |43.89     |43.22       |**45.08**
> **Table 1**: The results of using Vicuna-7B as the uncertainty measurement model and testing on Vicuna-7B.
>
> | LLaMA-2-7B-Chat-HF 	     | Truncate |BERT	 | Contriever| BGE-M3| BGE-Large| GRAGON-PLUS|Ours(Vicuna-7B)|
> |  ----  | :----:  | :----:  | :----:  | :----:  | :----:  | :----:  | :----:  |
> | 2WikiMultihopQA|28.50		|32.73   |33.60      |29.64  |34.14     |34.61       |**35.74**
> | Musique    	 | 9.41		|18.74   |14.25      |**24.27**  |24.20     |20.50       |22.68
> | TREC			 |64.50		|66.00   |70.00      |**71.00**  |70.50     |70.50       |67.50
> | Triviaqa		 |77.80		|78.69   |76.09      |75.74  |75.10     |77.51       |**79.46**
> | SAMSum		 |40.45		|40.01   |38.45      |40.37  |40.34     |41.08       |**42.70**
> | Average		 |44.13		|47.23   |46.48      |48.20  |48.85     |48.84       |**49.62**
> **Table 2**: The results of using Vicuna-7B as the uncertainty measurement model and testing on LLaMA-2-7B-Chat-HF.
>
>
>
> 1): We observe that UncertaintyRAG performs effectively in measuring uncertainty when used with other models.
>
> 2): Furthermore, we conduct RAG experiments with the model using a retrieval model trained on uncertainty measurements obtained from Vicuna-7B with LLaMA-2-7B-Chat-HF, and the results are presented in the table above.
>
> 3): We used different models (including Vicuna-7B) to estimate uncertainty, and observed consistent improvements in RAG experiments conducted on different models (Vicuna-7B, LLaMA-2-7B-Chat-HF), demonstrating that various models can achieve agreement in uncertainty measurement.

---

> ### Author Response · Authors · 2024-11-24
> **Response to Reviewer jHwd (Part 2/2)**
>
> **Question 2:** The evaluation tasks should include single-doc tasks such as NQ and Qasper. To still keep the distribution shifts, you can move these two tasks from training set to the evaluation set, and add the three few-shot learning tasks into the training data.
>
> **Response 2:**
>
> Thank you for your valuable suggestions. Based on your advice, we have reselected the training and test datasets. In the new experiments, the training set includes MultiFieldQA, Musique, QMSum, TREC, and TriviaQA, while the test set consists of NarrativeQA, Qasper, SAMSum, and 2WikiMultihopQA. The detailed experimental results are presented below:
>
> | LLaMA-2-7B-Chat-HF   | Truncate |BERT	 | Contriever| BGE-M3| BGE-Large| GRAGON-PLUS|Ours|
> |  ----  | :----:  | :----:  | :----:  | :----:  | :----:  | :----:  | :----:  |
> | 2WikiMultihopQA|28.50		| 32.73  |33.60		 |29.64  |34.14     |34.61       |**37.31**
> | NarrativeQA    |17.31		| 17.58  |16.91		 |19.45  |19.46     |20.01       |**20.20**
> | Qasper 		 |18.14		| 19.07  |**21.01**       |20.31  |19.93     |20.50       |20.81
> | SAMSum         |40.45		| 40.01  |38.45		 |40.37  |40.34     |41.08       |**41.26**
> | Average        |26.10		| 27.34  |27.49		 |27.44  |28.47     |29.05       |**29.90**
> **Table 3**: The results of adding NarrativeQA and Qasper to the evaluate datasets.
>
> All experiments were conducted on LLaMA-2-7B-Chat-HF. As observed, our method continues to achieve the best performance on the new datasets. This demonstrates the robustness of our method, as it does not rely on the specific selection of datasets.
>
>
> **Question 3**:
>
> There should be some comparisons with more baseline RAG methods, such as Self-RAG, FLARE, and RankRAG.
>
> **Response 3**:
>
> Thank you very much for your suggestion. However, we would like to first point out that our method is orthogonal to the RAG methods you mentioned. Specifically, using self-RAG or similar RAG approaches does not conflict with employing our trained retrieval model. Nonetheless, we greatly appreciate your reminder, and we have supplemented our experiments with results showing the performance of our method when combined with self-RAG.
>
> | self-RAG_llama2_7b |BERT	 | Contriever| BGE-M3| BGE-Large| GRAGON-PLUS|Ours|
> |  ----  | :----:  | :----:  | :----:  | :----:  | :----:  | :----:  |
> | 2WikiMultihopQA   |22.93   |19.94      |23.11  |22.91     |23.01       |**23.87**
> **Table 4**: The experimental results of combining self-RAG and UncertaintyRAG.
>
> 1):In this experiment, we used the official release model self-RAG_llama2_7b for testing. It is likely that the performance of the LLaMA-2-7B model is somewhat inferior to that of LLaMA-2-7B-Chat-HF, leading to a decline in overall experimental results compared to those obtained with LLaMA-2-7B-Chat-HF. Nonetheless, we were pleasantly surprised to find that our method still outperforms other baselines.
>
> 2):Specifically, the prompt used in this experiment is as follows:
>
>
> ```python
> def format_prompt(input, paragraph=None):
>   prompt = "### Instruction:\n{0}\n\n### Response:\n".format(input)
>   if paragraph is not None:
>     prompt += "[Retrieval]<paragraph>{0}</paragraph>".format(paragraph)
>   return prompt
>
> prompt = format_prompt(f"Answer the question based on the given passages.\nOnly give me the answer and do not output any other words.\n\nQuestion: {data[i]['input']}", prompt)
> ```
>
> If there are any questions, please let us know. And if you think that we have addressed your concerns, could you please consider raising the score? Thank you very much for your support.

---

> > ### Author Response · Authors · 2024-12-02
> > **Waiting for further discussion**
> >
> > Dear Reviewer jHwd,
> >
> > Thank you very much for your invaluable feedback on our paper. We have meticulously reviewed each of your points and endeavored to address them thoroughly. We would greatly appreciate it if you could review our responses and let us know if you have any additional comments regarding the paper or the rebuttal. We are eager to embrace all your critiques and integrate them into our work.
> >
> > If you think that we have addressed your concerns, could you please consider raising the score? Thank you very much for your support.
> >
> > Best regards,
> >
> > The Authors

---

> ### Author Response · Authors · 2024-12-03
>
> Dear Reviewer jHwd,
>
> In our most recent response, we addressed the concerns you raised previously. However, it is possible that for various reasons, our response may not have reached you. If you find that our previous answers have resolved your concerns, would you kindly consider raising your score? We greatly appreciate your support.
>
> Thank you very much.
>
> Best regards,
>
> The Authors

---

### Official Review · Reviewer_VFfy · 2024-10-28

**Soundness:** 1
**Presentation:** 1
**Contribution:** 1
**Rating:** 3
**Confidence:** 5

**Summary:**

The paper presents an unsupervised retrieval approach developed for long-context retrieval-augmented generation (RAG), i.e. when RAG is used as a solution for processing long contexts in LLMs, by chunking long contexts and retrieving chunks relevant to the user’s query. The proposed unsupervised retrieval approach is trained using only chunk data, without any query data, and without using any external labeling of query-chunk relevance. A high-level intuition of the proposed retrieval approach is to produce labeling for whether two text chunks are semantically similar or not, based on the log-likelihood output by an LLM for a concatenation of the two chunks.  The proposed approach is tested in domain shift settings.

In more details, the paper introduces a signal-to-noise ratio (SNR) which is computed over token probabilities in a window of text, i.e. an average probability of each token in a window, divided by a variance of these probabilities (formula 2). The authors find empirically that for any piece of text, the SNR tends to decrease to small values, starting from some position in the text (Figure 1). Based on this observation, the Span uncertainty (SU) of a given piece of text is calculated as an average probability of tokens after the found position (formula 3).  An ablation on the necessity of using this position cut-off is presented in Table 1 (two last columns).

To obtain labeled data for training a retriever, the documents are first split into chunks of length 300. Chunks themselves act as search queries (“anchor” chunks), and for each anchor chunk top-M closest chunks are retrieved using bm25. Then SU scores are calculated for each concatenation of anchor chunk - bm25-retrieved chunk. The SU scores are used to split a set of bm25-retrieved chunks into positive and negative samples. Due to the infeasibility of considering all chunks as anchor chunks, a clustering procedure is employed to select a representative subset of anchor chunks.

**Strengths:**

* An important problem of training retrievers for domain shift settings is considered.
* Related work is broad and detailed, listing several relevant research branches and representative works in each branch.

**Weaknesses:**

1. Text needs substantial rewriting: many parts of the text are unclear and hard to understand. Below I give examples of argumentations which are unclear or incorrect. I was only able to understand the proposed method after reading the corresponding section several times. In addition, the figures in the text are too small.
2. The motivation for the proposed unsupervised retriever is unclear. The authors argue that “the lack of labeled data to determine if (query, chunk) pairs are related poses significant limitations for training retrieval models in RAG systems.” (lines 62-64). However, there is plenty of widely used high-quality data nowadays for training retrievers, e.g. MS Marco dataset [1], and also of various QA datasets, which makes unclear the motivation for developing a retriever which does not use any queries during training. This is also unclear to me how a retriever trained without any query data could work in practice; an additional analysis of query representations would be helpful to explain it. _[addressed in the authors response]_
3. The proposed method brings only small and inconsistent improvements over baselines. This can be seen from Table 1. Furthermore, some rows in this Table highlight “Ours” in bold while it is not the highest score, e.g. for Vicuna-7B + TriviaQA and SamSum, or LLaMA-2-13B-Chat-HF and SamSum.
4. One missing simple baseline for a proposed method is to produce labeling for semantic similarity between two text chunks by simply prompting an LLM for this task. Furthermore, the paper does not cite nor compare to a very relevant body of work on using LLMs for reranking, e.g. [2] and [3].
5. The method is tested on five datasets: 2WikiMultihopQA, Musique, TREC, TriviaQA, and SAMSum. The paper does not provide any motivation for this selection _[motivation provided in the authors response]_. At the same time, in Information retrieval there are well established benchmarks for testing retrievers in domain shift settings, e.g. BEIR [4], which should be used for corresponding evaluations.
6. Experimental settings are given with very little detail, e.g. I was not able to find which metric is reported in the main Table 1, or what is the architecture of the proposed model (or what pretrained model it was initialised from). _[these two details were provided in the authors response]_. Other missing details include evaluation dataset sizes, how retrieval datastores were constructed, how many passages were retrieved per query, or which decoding strategy was used.

[1] Tri Nguyen et al. MS Marco: A Human Generated MAchine Reading COmprehension Dataset. NeurIPS 2016.

[2] Weiwei Sun et al. Is ChatGPT Good at Search? Investigating Large Language Models as Re-Ranking Agents. EMNLP 2023

[3] Zhen Qin et al. Large Language Models are Effective Text Rankers with Pairwise Ranking Prompting. NAACL 2024.

[4] Nandan Thakur et al. BEIR: A Heterogeneous Benchmark for Zero-shot Evaluation of Information Retrieval Models. NeurIPS Datasets and  Benchmarks track, 2021.


Examples of unclear parts of text (not an exhaustive list of all unclear parts).

Introduction:

Line 44: “these methods are generally difficult to achieve context length extrapolation.” -> an unclear phrase “methods are difficult to achieve”, and an unclear sentence why KV cache compression methods are poor in context length extrapolation.

Line 50: “Another lightweight solution for handling long contexts is to utilize long-context Retrieval-Augmented Generation (RAG) for long-context chunking”. -> long-context RAG is used _together_ with long-context chunking, not for it.

Lines 52-55: “Long context Retrieval-Augmented Generation refers to a method in natural language processing where a model retrieves relevant information from large external sources to assist in generating responses. It extends the traditional RAG by handling much longer input contexts.” -> The definition for  Long context RAG (sentence 1)  seems to be describing the general RAG setting. Sentence 2 is unclear: handling longer input contexts in the generator LLM or in the “ large external sources”?

Line 56: “ Typically, RAG employs existing LLMs with limited context windows to retrieve relevant chunks, either with or without semantic truncation. Retrieval models are commonly used for this purpose; however, they require a large amount of high-quality labeled data for training, which limits their scalability and adaptability. ” -> RAG does not employ LLMs to retrieve relevant chunks, this is done by the retriever. Term “semantic truncation” is not introduced nor explained. High-quality labeled data for training retrievers is available, e.g. the widely used MS Marco dataset [3].

Line 61: “Modern RAG systems rely on complex chunking methods (Sarthi et al., 2024)” -> this is an incorrect claim, most of modern RAG systems use the same simple strategy of chunking into the fixed number of tokens  or sentences, as is done in this submission.

Lines 64-65: “Recent research combines RAG with long-context LLMs to handle extended contexts and mitigate semantic incoherence in chunk processing” ->  I did not understand what “semantic incoherence in chunk processing” means.

Line 71: “Additionally, the complexity of these methods makes them vulnerable to distribution shifts” -> I did not understand how these two concepts are connected.

Lines 77-78: “we introduce a novel uncertainty estimation technique based on the Signal-to-Noise Ratio (SNR), which stabilizes predictions and reduces biases from random chunk splitting.” -> I did not understand what “stabilizes predictions” means (and why they are “unstable” in the first place), and what are the “biases from random chunk splitting”.

Related Work:

Line 131: “However, considering that many LLMs are closed-source, fine-tuning these models specifically for a retrieval model is impractical” -> (1) I did not understand what  “fine-tuning these models specifically for a retrieval model” means. (2) there are a lot of open source LLMs available nowadays.

Methodology:

Line 190: “In this section, we introduce a span uncertainty method based on SNR to obtain confidence scores between chunks” -> what does “confidence scores between chunks” mean?

Line 195: “This process typically does not involve using query data for training, so we further develop two methods to scale chunk data to enhance the model’s distribution shift generalization ability” -> I did not understand how the second part logically follows from the first one.

Line 244: “Additionally, we have observed a specific window in the model’s output log probabilities. The probability distribution within this window is stable, which we believe indicates better calibration” -> a “specific window” where? in the vocabulary dimension, or in the text dimension? what does it mean “the probability distribution is stable”? Why does it indicate better calibration?

Lines 293-294: “We set up two windows within M, each containing m samples. Among them, the top m are positive samples, and the bottom m are negative samples.” ->  How exactly are these windows of m samples set up? Furthermore, is the “top m” computed over bm 25 scores or $\hat S_ij$ scores?

Caption of Figure 1 is also hard to understand.

Regarding describing the methodology (Section 3), I would suggest showing all the steps on a particular text example. _[presented in the authors response]_

**Questions:**

1. What is the intuition of the proposed SNR? What does it mean qualitatively when $SNR_j < \sigma$? E.g. does it mean that a window of text is predicted well by an LLM, or the opposite?

2. Formula (3): when a piece of text is input to the LLM, it outputs probabilities of tokens conditions on _all_ their preceding tokens, till $x_0$. However, formula (3) uses probabilities conditioned only on tokens within a specific window, till $x_{0j}$. How these probabilities are computed?

_[both questions answered in the authors response]_

---

> ### Author Response · Authors · 2024-11-24
> **Response to Reviewer VFfy (Part 1/6)**
>
> Dear Reviewer VFfy,
>
> Thank you very much for your comment, we will address your concerns below.
>
>
> **Question 1**: What is the intuition of the proposed SNR? What does it mean qualitatively when SNRj<σ ? E.g. does it mean that a window of text is predicted well by an LLM, or the opposite?
>
>
>
>
> **Response 1**:
>
> Thank you for the insightful question.
>
> (1) In fact, the overall experiment is based on our observations presented in Figure 1. We noticed a sudden drop in the SNR before the end of the first chunk, followed by a stable plateau. This led us to hypothesize that this SNR behavior could reflect the stability of the model's uncertainty. Our subsequent experiments have confirmed this hypothesis.
>
> (2) Your understanding is correct: when $SNR_j < \sigma$, it indicates that the corresponding window of text is predicted well by the LLM. To help you better understand, we have uploaded an anonymous code file that provides a detailed example: https://ufile.io/56d9jxyh.
>
>
> **Question 2**: Formula (3): when a piece of text is input to the LLM, it outputs probabilities of tokens conditions on all their preceding tokens, till x0. However, formula (3) uses probabilities conditioned only on tokens within a specific window, till x0j. How these probabilities are computed?
>
>
>
>
> **Response 2**:
>
> Thank you for the insightful question.
> In fact, when we input a piece of text into an LLM, we can obtain the self-information(The lower the self-information, the higher its probability) for all tokens, which can be stored in a list. Subsequently, when calculating the SNR, we only use the self-information within a sliding window for computation.
>
>
> To help you better understand our work, we will next demonstrate all the steps involved in measuring uncertainty using a particular text example. We also sincerely appreciate your suggestion and will include this as a case study in the appendix to facilitate future readers' comprehension.
> Now, assume we have two chunks:
> - **Chunk A:** *How do you do?*
> - **Chunk B:** *Thank you for taking the time to review our manuscript.*
>
>
>
>
> Here, for the sake of convenience, we treat each word or punctuation mark as a token.
>
>
>
>
> In this case, Chunk A consists of 5 tokens, and we can obtain the self-information for these 5 tokens. Similarly, Chunk B consists of 11 tokens, for which we also have their self-information. By concatenating these self-information, we get:
> $
> P_{A1}, P_{A2}, \dots, P_{A5}, P_{B1}, P_{B2}, \dots, P_{B11}
> $.
>
>
> Assume the sliding window size is set to 3, with a stride of 1 (used here for clearer demonstration). We first calculate the SNR within the window:
> $
> (P_{B9}, P_{B10}, P_{B11})
> $
> and find that it is less than the set threshold.
>
>
> We then slide the window one step to the left and calculate:
> $
> (P_{B8}, P_{B9}, P_{B10})
> $
> and continue this process iteratively.
>
>
>
>
> For this example, let us assume we eventually find that the SNR for:
> $
> (P_{A4}, P_{A5}, P_{B1})
> $
> exceeds the threshold. At this point, the sliding window stops.
>
>
>
>
> We then take the self-information of all tokens from the previous sliding windows (i.e.,  $\( P_{A5} \)$ to $\( P_{B11} \)$) and compute their average. This average value is used as a similarity score between the two chunks, forming positive and negative samples for subsequent contrastive learning training.

---

> > ### Author Response · Authors · 2024-11-25
> > **REFERENCES**
> >
> > [1] Peng B, Quesnelle J, Fan H, et al. Yarn: Efficient context window extension of large language models. arXiv preprint arXiv:2309.00071, 2023.
> >
> >
> > [2] Chen G, Li X, Meng Z, et al. Clex: Continuous length extrapolation for large language models. arXiv preprint arXiv:2310.16450, 2023.
> >
> >
> > [3] Ding Y, Zhang L L, Zhang C, et al. Longrope: Extending llm context window beyond 2 million tokens. arXiv preprint arXiv:2402.13753, 2024.
> >
> >
> > [4] Gautier Izacard,Mathilde Caron,Lucas Hosseini,Sebastian Riedel,Piotr Bojanowski,Armand Joulin,and Edouard Grave.Unsupervised dense information retrieval with contrastive learning. arXiv preprint arXiv:2112.09118,2021.
> >
> >
> > [5] Jianlv Chen,Shitao Xiao,Peitian Zhang,Kun Luo,Defu Lian,and Zheng Liu.Bgem3-embedding: Multi-lingual,multi-functionality,multi-granularity text embeddings through self-knowledge distillation.arXiv preprint arXiv:2402.03216,2024.
> >
> >
> > [6] Sheng-Chieh Lin,Akari Asai,Minghan Li,Barlas Oguz,Jimmy Lin,Yashar Mehdad,Wen-tau Yih, and Xilun Chen.How to train your dragon: Diverse augmentation towards generalizable dense retrieval.arXiv preprint arXiv:2302.07452,2023a.
> >
> >
> > [7] Kun Luo,Zheng Liu,Shitao Xiao, and Kang Liu. Bge landmark embedding:A chunking-free embedding method for retrieval augmented long-context large language models. arXiv preprint arXiv:2402.11573,2024.
> >
> >
> > [8] Ohad Rubin, Jonathan Herzig, and Jonathan Berant. Learning to retrieve prompts for in-context learning. arXiv preprint arXiv:2112.08633, 2021.
> >
> >
> > [9] Jiacheng Ye, Zhiyong Wu, Jiangtao Feng, Tao Yu, and Lingpeng Kong. Compositional exemplars for in-context learning. arXiv preprint arXiv:2302.05698, 2023.
> >
> >
> > [10] Yushi Bai,Xin Lv, Jiajie Zhang,Hongchang Lyu, Jiankai Tang,Zhidian Huang,Zhengxiao Du, Xiao Liu,Aohan Zeng,Lei Hou,Yuxiao Dong, Jie Tang , and Juanz iLi.Longbench:Abilingual, multitask benchmark for long context understanding.arXiv preprint arXiv:2308.14508,2023.

---

> ### Author Response · Authors · 2024-11-25
> **Response to Reviewer VFfy (Part 2/6)**
>
> **Weakness**
>
> **Weakness 1** Text needs substantial rewriting: many parts of the text are unclear and hard to understand. Below I give examples of argumentations which are unclear or incorrect. I was only able to understand the proposed method after reading the corresponding section several times. In addition, the figures in the text are too small.
>
>
> **Response to weakness 1**: Thank you very much for your suggestions regarding the writing details in our paper. We will address the issues you mentioned one by one in the following responses.
>
>
> Introduction:
> Line 44: “these methods are generally difficult to achieve context length extrapolation.” -> an unclear phrase “methods are difficult to achieve”, and an unclear sentence why KV cache compression methods are poor in context length extrapolation.
>
>
> **Response**: Thank you for the suggestion. In fact, the KV cache compression method is a way to reduce memory cost, but not related to the model length extrapolation performance. And usually,  the context length extrapolation is usually poor, based on previous works[1][2][3].
>
>
>
>
> Line 50: “Another lightweight solution for handling long contexts is to utilize long-context Retrieval-Augmented Generation (RAG) for long-context chunking”. -> long-context RAG is used together with long-context chunking, not for it.
>
>
> **Response**: Thank you for the suggestion. Here's the revised version of the sentence
> based on the feedback:
> "Another lightweight solution for handling long contexts is to utilize long-context Retrieval-Augmented Generation (RAG) in conjunction with long-context chunking."
>
>
> Lines 52-55: “Long context Retrieval-Augmented Generation refers to a method in natural language processing where a model retrieves relevant information from large external sources to assist in generating responses. It extends the traditional RAG by handling much longer input contexts.” -> The definition for Long context RAG (sentence 1) seems to be describing the general RAG setting. Sentence 2 is unclear: handling longer input contexts in the generator LLM or in the “ large external sources”?
>
> **Response**:  Thank you very mush for the feedback. Here's the revised version of the paragraph to address the concerns:
> "Long-context Retrieval-Augmented Generation (RAG) refers to a method in natural language processing that addresses the long-context problem by retrieving relevant information from large external sources. This process helps LLMs effectively access more useful information within their limited context window, enabling them to handle much longer input contexts indirectly."
>
>
> Line 56: “ Typically, RAG employs existing LLMs with limited context windows to retrieve relevant chunks, either with or without semantic truncation. Retrieval models are commonly used for this purpose; however, they require a large amount of high-quality labeled data for training, which limits their scalability and adaptability. ” -> RAG does not employ LLMs to retrieve relevant chunks, this is done by the retriever. Term “semantic truncation” is not introduced nor explained. High-quality labeled data for training retrievers is available, e.g. the widely used MS Marco dataset [3].
>
> **Response**:
> (1) Thank you very much for your notice,  we revised the presentation to "Typically, RAG helps existing LLMs with limited context windows process retrieved relevant chunks. These chunks are obtained by retrieval models, either with or without addressing semantic truncation, which refers to the phenomenon where splitting chunks results in semantic discontinuity between adjacent chunks."
>
>
> (2) While it is true that high-quality labeled datasets like MS MARCO are available and widely used, it is also important to note that the vast majority of textual data remains unlabeled. Recent works, such as Contriever[4], BGE-M3[5], Dragon[6] and other methods trained on unlabeled data, have demonstrated significant progress in leveraging these abundant resources. These efforts aim to utilize the vast scale of unlabeled data effectively, as they far outweigh the availability of labeled datasets. This growing focus on unsupervised and self-supervised learning highlights the potential of using unlabeled data for training retrievers, providing broader applicability and scalability in real-world scenarios.

---

> ### Author Response · Authors · 2024-11-25
> **Response to Reviewer VFfy (Part 3/6)**
>
> Line 61: “Modern RAG systems rely on complex chunking methods (Sarthi et al., 2024)” -> this is an incorrect claim, most of modern RAG systems use the same simple strategy of chunking into the fixed number of tokens or sentences, as is done in this submission.
> And
> Lines 64-65: “Recent research combines RAG with long-context LLMs to handle extended contexts and mitigate semantic incoherence in chunk processing” -> I did not understand what “semantic incoherence in chunk processing” means.
>
> **Response**: (1) In fact, some recent studies have shown that using the simple strategy of chunking can disrupt the coherence of the context, thereby affecting the quality of the embeddings [7]. We also revise the presentation: Modern RAG systems may rely on complex chunking methods (Sarthi et al., 2024)
>
> (2) Semantic incoherence in chunk processing refers to a situation where a coherent segment of information is fragmented into different chunks before its meaning is fully conveyed. For example, consider a Chain of Thought for solving a mathematical problem: if the argument is interrupted and divided into separate chunks during the explanation, it naturally leads to semantic incoherence.
>
>
>
>
> Line 71: “Additionally, the complexity of these methods makes them vulnerable to distribution shifts” -> I did not understand how these two concepts are connected.
>
> **Response**: As mentioned earlier, some methods require fine-tuning adapters for specific large language models (LLMs) to manage chunk representation compression. In such cases, these methods are naturally susceptible to changes in the distribution of the LLMs or the data.
>
>
>
>
>
>
> Lines 77-78: “we introduce a novel uncertainty estimation technique based on the Signal-to-Noise Ratio (SNR), which stabilizes predictions and reduces biases from random chunk splitting.” -> I did not understand what “stabilizes predictions” means (and why they are “unstable” in the first place), and what are the “biases from random chunk splitting”.
>
> **Response**:
> (1)To help you better understand, we have uploaded an anonymous code file that provides a detailed example: https://ufile.io/56d9jxyh. It can be observed that the self-information of the initial tokens is very high, indicating instability in the model's predictions. By using the SNR-based calibration method, we can effectively identify the point where the self-information stabilizes at a relatively low level. Subsequent experiments further demonstrate that using the self-information of these tokens to estimate the similarity between two chunks achieves better performance.
>
>
> (2)  “biases from random chunk splitting” refers to the phenomenon of semantic incoherence caused by random splitting.
>
>
> Related Work:
>
> Line 131: “However, considering that many LLMs are closed-source, fine-tuning these models specifically for a retrieval model is impractical” -> (1) I did not understand what “fine-tuning these models specifically for a retrieval model” means. (2) there are a lot of open source LLMs available nowadays.
>
> **Response**:
> (1) Thank you very much for your suggestion. We have revised the sentence to: “However, considering that many LLMs are closed-source, end-to-end training of LLMs with retrievers is impractical.”
>
> (2) On one hand, closed-source models cannot be trained end-to-end, and many of them do not provide access to logits. On the other hand, while there are many open-source models available, we aim to make our model adaptable to all types of models.
> Methodology:
>
> Line 190: “In this section, we introduce a span uncertainty method based on SNR to obtain confidence scores between chunks” -> what does “confidence scores between chunks” mean?
>
> **Response**: Thank you for the feedback. The expression should be revised to accurately reflect 'similarity between chunks' instead.
>
> Line 244: “Additionally, we have observed a specific window in the model’s output log probabilities. The probability distribution within this window is stable, which we believe indicates better calibration” -> a “specific window” where? in the vocabulary dimension, or in the text dimension? what does it mean “the probability distribution is stable”? Why does it indicate better calibration?
>
> **Response**:
> (1) In the text dimension, the detailed process is presented in our response to Question 2.
> (2) For further details, please refer to the response provided to Lines 77-78. What’s more, our subsequent experiments consistently demonstrate the effectiveness of our method.

---

> ### Author Response · Authors · 2024-11-25
> **Response to Reviewer VFfy (Part 4/6)**
>
> Lines 293-294: “We set up two windows within M, each containing m samples. Among them, the top m are positive samples, and the bottom m are negative samples.” -> How exactly are these windows of m samples set up? Furthermore, is the “top m” computed over bm 25 scores or S^ij scores?
>
> **Response**:
> (1) Thank you for your valuable feedback, here we followed the settings from previous works[8][9].
>
> (2) It’s computed over S^ij scores. To clarify, we have revised the statement as follows: "We set up two windows within M, each containing m samples. These samples are ranked by S^ij scores, with the top m considered positive samples and the bottom m considered negative samples."
>
>
> Regarding describing the methodology (Section 3), I would suggest showing all the steps on a particular text example.
> **Response**: Thank you for the insightful suggestion. the detailed process is presented in our response to Question 2.
>
>
>
>
>
>
> **Weakness 2**The motivation for the proposed unsupervised retriever is unclear. The authors argue that “the lack of labeled data to determine if (query, chunk) pairs are related poses significant limitations for training retrieval models in RAG systems.” (lines 62-64). However, there is plenty of widely used high-quality data nowadays for training retrievers, e.g. MS Marco dataset [1], and also of various QA datasets, which makes unclear the motivation for developing a retriever which does not use any queries during training. This is also unclear to me how a retriever trained without any query data could work in practice; an additional analysis of query representations would be helpful to explain it.
>
>
> **Response to weakness 2**: Thank you for the question. In fact, many influential works, such as Contriever[4], BGE-M3[5], and Dragon[6], are attempting to utilize unlabeled data. We have specifically addressed this point in detail in our response to Weakness 1 regarding line 56.
>
>
> **Weakness 3**The proposed method brings only small and inconsistent improvements over baselines. This can be seen from Table 1. Furthermore, some rows in this Table highlight “Ours” in bold while it is not the highest score, e.g. for Vicuna-7B + TriviaQA and SamSum, or LLaMA-2-13B-Chat-HF and SamSum.
>
>
> **Response to weakness 3**:
> (1) Compared to other baselines, our approach achieves better performance using significantly less data, and we also demonstrate the potential for scaling up. This indicates that with more computational resources and data, our method has the potential to achieve even greater improvements.
>
> (2) **Additionally, we would like to point out that even when comparing two baselines (e.g., Contriever and GRAGON-PLUS), the performance differences vary across different LLMs.** For instance, on LLaMA-2-7B-Chat-HF, GRAGON-PLUS outperforms Contriever by an average of **2.36%**; on LLaMA-2-13B-Chat-HF, this advantage decreases to **0.78%**; and on Vicuna-7B, it further drops to **0.39%**. This suggests that the gains achieved by retrieval models differ across LLMs. Moreover, the improvement of our model over the baselines is already substantial compared to the differences between the baselines themselves.
>
> (3) Thank you very much for your reminder! We will correct the mistakenly highlighted portions in the table.

---

> ### Author Response · Authors · 2024-11-25
> **Response to Reviewer VFfy (Part 5/6)**
>
> **Weakness 4** One missing simple baseline for a proposed method is to produce labeling for semantic similarity between two text chunks by simply prompting an LLM for this task. Furthermore, the paper does not cite nor compare to a very relevant body of work on using LLMs for reranking, e.g. [2] and [3].
>
>
> **Response to weakness 4**: Thank you for your suggestion. However, on one hand, using LLM models for re-ranking involves significantly higher computational costs compared to using retrievers. On the other hand, we have conducted some experiments and found that when dealing with extremely long texts, which need to be divided into numerous chunks, directly employing LLMs for re-ranking through prompts does not achieve satisfactory results. Below, we will present the experimental results of RankGPT along with the corresponding code.
>
>
> | Vicuna-7B-16k    | Truncate |BERT	 | Contriever| BGE-M3| BGE-Large| GRAGON-PLUS|RankGPT| Ours
> |  ----  | :----:  | :----:  | :----:  | :----:  | :----:  | :----:  | :----:  | :----:  |
> | 2WikiMultihopQA|23.15		|27.80   |28.33      |27.47  |29.50     |29.51       |21.72 |29.88
> | Musique    	 | 8.11		|13.12   | 9.88      |13.02  |14.15     |13.24       | 7.32 |14.94
> | TREC			 |66.50		|67.50   |69.50      |69.00  |69.00     |70.00       |60.50 |68.50
> | Triviaqa		 |83.96		|82.93   |85.74      |84.46  |84.56     |83.05       |84.42 |84.98
> | SAMSum		 | 7.80		|20.75   |19.88      |19.50  |19.32     |20.67       |16.14 |19.77
> | Average		 |37.91		|42.42   |42.51      |42.68  |43.30     |43.29       |38.03 |43.62
> **Table 1:** Results of RankGPT on Vicuna-7B-16k
>
>
> **Code**
>
> ```python
> for i in range(len(data)):
>     d_doc = text_splitter.split_text(data[i]['context'])
>     hits = [{'content':d} for d in d_doc]
>     item = {'query':data[i]['input'], 'hits':hits}
>     new_item = permutation_pipeline(item, rank_start=0, rank_end=len(d_doc)-1, model_name=model_name, llm=llm)
>     prompt = "\n".join([h['content'] for h in new_item['hits'][:30]])
>
>     prompt = f"Answer the question based on the given passages. Only give me the answer and do not output any other words.\n\nThe following are given passages.\n{prompt}\n\nAnswer the question based on the given passages. Only give me the answer and do not output any other words.\n\nQuestion: {data[i]['input']}\nAnswer:",
>
>     completions = llm.generate(prompt, sampling_params)
>     outputs.append(completions[0].outputs[0].text)
> ```
>
> The permutation_pipeline function is based on the official code from RankGPT, available at https://github.com/sunnweiwei/RankGPT/blob/main/rank_gpt.py. We modified the run_llm function within it to include an option for using open-source models.
>
> ```python
> def run_llm(messages, api_key=None, model_name="gpt-3.5-turbo", llm=None):
>     if 'gpt' in model_name:
>         Client = OpenaiClient
>     elif 'claude' in model_name:
>         Client = ClaudeClient
>     elif 'llama' in model_name or 'vicuna' in model_name:
>         messages = "\n".join([f"{entry['role'].capitalize()}: {entry['content']}" for entry in messages])
>         from vllm import LLM, SamplingParams
>
>         stop_tokens = ["Question:", "Question", "USER:", "USER", "ASSISTANT:", "ASSISTANT", "Instruction:", "Instruction", "Response:", "Response"]
>         sampling_params = SamplingParams(temperature=0.0, top_p=1, max_tokens=512, stop=stop_tokens)
>         response = llm.generate(messages, sampling_params)[0].outputs[0].text
>         return response
>     else:
>         Client = LitellmClient
>
>     agent = Client(api_key)
>     response = agent.chat(model=model_name, messages=messages, temperature=0, return_text=True)
>     return response
>
> ```

---

> ### Author Response · Authors · 2024-11-25
> **Response to Reviewer VFfy (Part 6/6)**
>
> **Weakness 5** The method is tested on five datasets: 2WikiMultihopQA, Musique, TREC, TriviaQA, and SAMSum. The paper does not provide any motivation for this selection. At the same time, in Information retrieval there are well established benchmarks for testing retrievers in domain shift settings, e.g. BEIR [4], which should be used for corresponding evaluations.
>
>
> **Response to weakness 5**: Thank you for the insightful question. In fact, we carefully considered the diversity of both the training and test datasets when making our selections. Specifically:
>
> 2WikiMultihopQA and Musique are tasks that require answering questions based on multiple provided documents.
>
> TREC is a classification task that involves categorizing questions into 50 predefined categories.
>
> TriviaQA is a single-document question-answering task with several few-shot examples provided.
>
> SAMSum is a dialogue summarization task, also including few-shot examples.
>
> To address your concerns, we have reselected the training and test datasets. In the new experiments, the training set includes MultiFieldQA, Musique, QMSum, TREC, and TriviaQA, while the test set consists of NarrativeQA, Qasper, SAMSum, and 2WikiMultihopQA. The detailed experimental results are presented below.
>
> | LLaMA-2-7B-Chat-HF   | Truncate |BERT	 | Contriever| BGE-M3| BGE-Large| GRAGON-PLUS|Ours|
> |  ----  | :----:  | :----:  | :----:  | :----:  | :----:  | :----:  | :----:  |
> | 2WikiMultihopQA|28.50		| 32.73  |33.60		 |29.64  |34.14     |34.61       |**37.31**
> | NarrativeQA    |17.31		| 17.58  |16.91		 |19.45  |19.46     |20.01       |**20.20**
> | Qasper 		 |18.14		| 19.07  |**21.01**       |20.31  |19.93     |20.50       |20.81
> | SAMSum         |40.45		| 40.01  |38.45		 |40.37  |40.34     |41.08       |**41.26**
> | Average        |26.10		| 27.34  |27.49		 |27.44  |28.47     |29.05       |**29.90**
> Table 2: The results of new experiments
>
> All experiments were conducted on LLaMA-2-7B-Chat-HF. As observed, our method continues to achieve the best performance on the new datasets. This demonstrates the robustness of our method, as it does not rely on the specific selection of datasets.
>
>
>
> **Weakness 6**Experimental settings are given with very little detail, e.g. I was not able to find which metric is reported in the main Table 1, or what is the architecture of the proposed model (or what pretrained model it was initialised from). Other missing details include evaluation dataset sizes, how retrieval datastores were constructed, how many passages were retrieved per query, or which decoding strategy was used.
>
>
> **Response to weakness 6**:  Thanks for your suggestion! Regarding the metrics, we followed the choices made by LongBench [7]. Specifically, for 2WikiMultihopQA, Musique, and TriviaQA, we used the F1-score, for TREC, we used accuracy, and for SAMSum we used the ROUGE score. We initialize the model and configure additional settings based on EPR[8]. We will add more detailed explanations to the paper to reduce any potential confusion for readers. Once again, thank you for your reminder!
>
> **If there are any questions, please let us know. And if you think that we have addressed your concerns, could you please consider raising the score? Thank you very much for your support.**

---

> ### Author Response · Authors · 2024-12-03
>
> Dear Reviewer VFfy,
>
> In our most recent response, we addressed the concerns you raised previously. However, it is possible that for various reasons, our response may not have reached you. If you find that our previous answers have resolved your concerns, would you kindly consider raising your score? We greatly appreciate your support.
>
> Thank you very much.
>
> Best regards,
>
> The Authors

---

> > ### Comment · Reviewer_VFfy · 2024-12-03
> >
> > Thank you for updating the results!
> >
> > > The latest version can be accessed via the anonymous link
> >
> > I believe providing the updated pdf is not allowed at this stage of the author-discussion process. I would also like to note that since the original response from the authors was posted in the end of the initial author-reviewer discussion schedule, and was time consuming to read (6 pages), I was not able to post my previous response earlier.
> >
> > > New results
> >
> > Thank you for providing new baselines according to my comments. I would like to note that in these results, for 3 out of 5 datasets, Ours is very close to the baseline labelers (68.0 vs 67.5, 80.41 vs 80.21, 42.49 vs 42.51).
> >
> > This also connects to my initial Weakness 5 about the need for testing the proposed approach on more (well-established) datasets.
> >
> > > This could potentially be an area for future research—specifically, how to use listwise LLM reranking for scoring. However, this is beyond the scope of our current work.
> >
> > I am not sure. There are recent more advanced approaches for LLM reranking, e.g. [3] in my original review, and when i searched for the open implementation, I found also e.g. this library (https://github.com/ielab/llm-rankers) which implements various advanced LLM-based approaches. The main novelty of the paper is the way of ranking documents (using an LLM) with the proposed SNR, so all these works seem as natural baselines.  Furthermore, from the current results we see that they are in fact often performing similarly to the proposed approach.
> >
> > I appreciate the authors efforts in addressing a part of my concerns, however I believe this work requires substantial changes beyond the rebuttal, including rewriting (cc W1), testing on more datasets (cc W5), and considering strong baselines for oracle labeling (cc W4).

---

### Official Review · Reviewer_rEbW · 2024-11-02

**Soundness:** 3
**Presentation:** 2
**Contribution:** 2
**Rating:** 6
**Confidence:** 3

**Summary:**

This paper presents UncertaintyRAG, a novel method for improving long-context Retrieval-Augmented Generation (RAG) by utilizing span-level Signal-to-Noise Ratio (SNR)-based uncertainty for estimating the similarity between text chunks. The authors propose an unsupervised learning technique for training the retrieval model, coupled with an efficient data sampling and scaling strategy. The key insight is that the SNR-based span uncertainty can be applied to chunk similarity estimation. The method achieves state-of-the-art performance on LLaMA-2-7B, with a 2.03% improvement over baselines using only 4% of the training data. The authors also emphasize that UncertaintyRAG is lightweight and can be seamlessly integrated into any large language model without the need for fine-tuning.

**Strengths:**

1. The focus on SNR-based span uncertainty to enhance retrieval in long-context RAG tasks is novel. Uncertainty quantification has been explored in other domains, but its application in chunk similarity estimation for RAG tasks is a fresh and valuable contribution.
2. The authors provide a solid methodology for both retrieval model training and data sampling, with clear explanations of how span uncertainty is calculated and used to construct positive and negative samples.
3. The experiments demonstrates how span-level uncertainty improves performance across multiple datasets, including distribution shift settings where traditional models often struggle.

**Weaknesses:**

1. While the paper introduces SNR-based span uncertainty as a novel metric for estimating text chunk similarity, it fails to provide a compelling motivation for why SNR is the most appropriate choice for this task. The choice of SNR seems somewhat ad hoc, and the paper would benefit from a comparative study showing why SNR outperforms these other potential metrics in various retrieval scenarios.
2. The paper relies on a relatively simple fixed chunk size (e.g., 300 words per chunk) for dividing long-context documents. However, chunking is a critical aspect of Retrieval-Augmented Generation (RAG) systems, and the paper does not explore more sophisticated chunking methods, such as dynamic or semantic-based chunking. This is a significant limitation because fixed-length chunking can lead to semantic fragmentation, where important contextual information is split across chunks, reducing retrieval accuracy.

**Questions:**

1. Can you provide more detailed justification for why Signal-to-Noise Ratio (SNR) was chosen as the primary uncertainty metric for span-level similarity estimation? How does it compare to other uncertainty metrics？
2. How does your method handle ambiguous queries where there may be multiple valid retrievals or interpretations? Does the SNR-based uncertainty provide any benefit in disambiguating between competing chunks of text?

---

> ### Author Response · Authors · 2024-11-25
> **Response to Reviewer rEbW (Part 1/2)**
>
> Dear Reviewer rEbW,
>
>
> Thank you very much for your comment, we will address your concerns below.
>
> **Question 1**: Can you provide more detailed justification for why Signal-to-Noise Ratio (SNR) was chosen as the primary uncertainty metric for span-level similarity estimation? How does it compare to other uncertainty metrics？
>
>
> **Response 1**:
> Thank you very much for your suggestion!
>
> (1) In fact, the overall experiment is based on our observations presented in Figure 1. We noticed a sudden drop in the SNR before the end of the first chunk, followed by a stable plateau. This led us to hypothesize that this SNR behavior could reflect the stability of the model's uncertainty. Our subsequent experiments have confirmed this hypothesis. To facilitate your understanding, we have uploaded an anonymized code example that illustrates a specific case: https://ufile.io/56d9jxyh
>
> (2) Table 4 in the original text provides a comparison of our method with other uncertainty metrics. The specific formulas for these uncertainty metrics are detailed in Appendix F. We directly replicate the experiments table here.
>
>
> | Method 	     	| 2WikiMultihopQA |Musique	 |  TREC| TriviaQA|  SAMSum| Average|
> |  ----  | :----:  | :----:  | :----:  | :----:  | :----:  | :----:  |
> | Minimum		 	|28.04		|15.25   |66.50      |77.21  |40.35     |45.47       |
> | Average    	 	|31.55		|17.72   |66.50      |78.21  |41.56     |47.10       |
> | Log-sum			|29.77		|16.25   |69.50      |80.21  |41.55     |47.25       |
> | Entropy		 	|31.95		|16.17   |69.00      |79.70  |40.99     |47.56       |
> | self-information	|34.60		|16.53   |65.50      |79.84  |41.32     |47.56       |
> | Ours		 	 	|37.27		|23.03   |68.00      |80.41  |42.49     |50.23       |
> **Table 1:** Comparison results of different uncertainty modeling approaches.
>
>
>
> **Question 2**:  How does your method handle ambiguous queries where there may be multiple valid retrievals or interpretations? Does the SNR-based uncertainty provide any benefit in disambiguating between competing chunks of text?
>
>
> **Response 2**:  Thank you for the insightful question. This is a very interesting idea. We designed the following experiment: first, we use BERT for coarse ranking to select a set of candidate chunks. Then, these candidate chunks are concatenated with the question and input into an LLM. Subsequently, we apply our method to measure the similarity scores between the question and the candidate chunks. Finally, the chunks are re-ranked based on these similarity scores. The detailed experimental results can be found in the table below:
>
>
>
>
> | LLaMA-2-7B-Chat-HF |BERT	 |  After re-ranking|
> |  ----  | :----:  | :----:  |
> | 2WikiMultihopQA| 32.73  |33.73		 |
> | Musique    	 | 18.74  |20.34		 |
> | TREC 		 	 | 66.00  |66.50		 |
> | Triviaqa       | 78.69  |79.78		 |
> | SAMSum         | 40.01  |41.24		 |
> | Average        | 47.23  |48.32		 |
> **Table 2:** Results after re-ranking
>
>
> We can find that after re-ranking, the final results show a significant improvement. This could potentially be an area for further exploration in the future. Once again, thank you for your valuable suggestion!

---

> ### Author Response · Authors · 2024-11-25
> **Response to Reviewer rEbW (Part 2/2)**
>
> **Weakness**
>
>
> **Weakness 1**: While the paper introduces SNR-based span uncertainty as a novel metric for estimating text chunk similarity, it fails to provide a compelling motivation for why SNR is the most appropriate choice for this task. The choice of SNR seems somewhat ad hoc, and the paper would benefit from a comparative study showing why SNR outperforms these other potential metrics in various retrieval scenarios.
>
>
> **Response to weakness 1**: Thank you for your question. For detailed information, please refer to the response to Question 1.
>
>
> **Weakness 2**: The paper relies on a relatively simple fixed chunk size (e.g., 300 words per chunk) for dividing long-context documents. However, chunking is a critical aspect of Retrieval-Augmented Generation (RAG) systems, and the paper does not explore more sophisticated chunking methods, such as dynamic or semantic-based chunking. This is a significant limitation because fixed-length chunking can lead to semantic fragmentation, where important contextual information is split across chunks, reducing retrieval accuracy.
>
>
> **Response to weakness 2**: In the original test set, for few-shot tasks such as TREC, TriviaQA, and SAMSum, the nature of these tasks requires full sentences to be presented in the prompt. Therefore, during retrieval for these three tasks, we used complete sentences. And it can be observed that our method also achieves excellent performance when using complete sentences.
>
> **If there are any questions, please let us know. And if you think that we have addressed your concerns, could you please consider raising the score? Thank you very much for your support.**

---

> ### Author Response · Authors · 2024-12-03
>
> Dear Reviewer rEbW,
>
> In our most recent response, we addressed the concerns you raised previously. However, it is possible that for various reasons, our response may not have reached you. If you find that our previous answers have resolved your concerns, would you kindly consider raising your score? We greatly appreciate your support.
>
> Thank you very much.
>
> Best regards,
>
> The Authors

---

### Official Review · Reviewer_Q3d1 · 2024-11-03

**Soundness:** 2
**Presentation:** 2
**Contribution:** 3
**Rating:** 5
**Confidence:** 3

**Summary:**

The paper introduces UncertaintyRAG, a novel long-context RAG framework that incorporates span-level uncertainty using the Signal-to-Noise Ratio (SNR) for similarity estimation between text chunks. This technique aims to improve calibration, reduce semantic inconsistencies from random chunking, and enhance model robustness under distribution shifts. The authors also propose an efficient unsupervised learning approach for training retrieval models with minimal labeled data, achieving state-of-the-art performance with a lightweight design that can integrate with various LLMs without fine-tuning.

**Strengths:**

1. The idea of using an SNR-based span uncertainty metric to calibrate the retrieval model is novel. Specifically, the method defines a chunk similarity metric based on the probabilities of the downstream decoder LLM, which can be used to train a retrieval model with contrastive learning.
2. The proposed method achieves better performance than existing retrieval models under distributional shifts in the context, proving it to be an easy-to-adapt retriever for a specific model.
3. The method can be trained on an automatically created dataset using self-supervised learning, reducing the need for manually labeled data.

**Weaknesses:**

1. Some content is difficult to understand. For example, in line 245, "The probability distribution within this window is stable," which does not seem to imply a stable distribution. Additionally, in Section 4.1, the phrase "number of chunks" is unclear, as the previous mention of the same term refers to the total number of chunks in the training set.
2. The definition of the negative samples is questionable. It appears that the negative samples are among the top $M$ most similar chunks to the anchor. If this is the method for selecting hard negative examples, the hyperparameters in the paper would play a crucial role.
3. Numerous hyperparameters are introduced, such as the threshold $\sigma$, the number of samples $M$ and $m$, the chunk size, and the sliding window size. However, the authors do not explain how these hyperparameters were chosen or how sensitive the model is to these values.
4. It is challenging to conclude that the method can be integrated with any LLMs, given that uncertainty is only calculated using Llama 2 7B. The tested models also use Llama 2 as their base, which shares the same pretraining data. From Table 1, it is apparent that when the uncertainty score model does not match the LLM, the performance gain from the proposed method decreases. This raises doubts about whether the retriever must match the LLM used.

**Questions:**

1. The chunks are divided based on a specific word length (300 in the paper). Below are questions regarding this design:
(1) It is possible for 300 words to correspond to more than 512 tokens for BERT. What would be your solution in this scenario?
(2) Do you consider punctuation as separators for words as well?
(3) If a chunk is cut mid-sentence, concatenating two "incomplete" sentences (chunks) may lead to a "sudden transfer" between them. This could affect uncertainty calculation. What are your thoughts on this?
2. What is the role of $N$ as defined in line 285?
3. How did you select the training and test sets from the LongBench suite?
4. Not all training and test sets feature distributional shifts in their contexts. For example, both HotpotQA and 2WikiMQA use Wikipedia as their source of context.
5. In line 451, how do you conclude that "RSA is an effective metric for assessing distribution shifts," given that you only have two data points (GSM8K and 2WikiMQA)?
6. The "10" mentioned in line 310 should be $k$.

---

> ### Author Response · Authors · 2024-11-24
> **Response to Reviewer Q3d1 (Part 1/3)**
>
> Dear Reviewer Q3d1,
>
> Thank you very much for your comment, we will address your concerns below.
>
>
> **Question 1**: The chunks are divided based on a specific word length (300 in the paper). Below are questions regarding this design: (1) It is possible for 300 words to correspond to more than 512 tokens for BERT. What would be your solution in this scenario? (2) Do you consider punctuation as separators for words as well? (3) If a chunk is cut mid-sentence, concatenating two "incomplete" sentences (chunks) may lead to a "sudden transfer" between them. This could affect uncertainty calculation. What are your thoughts on this?
>
>
> **Response 1**:
> (1) and (2): Thank you very much for your reminder. Our explanation in the paper was not sufficiently clear. We actually used 300 letters rather than words. The specific code is shown below.
> ```python
> text_splitter = CharacterTextSplitter(
>     separator="",
>     chunk_size=300,
>     chunk_overlap=0,
>     length_function=len,
>     is_separator_regex=False,
> )
> ```
>
>
> (3): In the original test set, for few-shot tasks such as TREC, TriviaQA, and SAMSum, the nature of these tasks requires full sentences to be presented in the prompt. Therefore, during retrieval for these three tasks, we used complete sentences. And it can be observed that our method also achieves excellent performance when using complete sentences.
>
> What's more, one of the primary objectives of our method is to address the fragmentation between chunks. To amplify the impact of this challenge, we allowed "mid-sentence" cases to exist and did not apply more sophisticated processing to the chunks.
> If only the output uncertainty is used for estimation (i.e., the PreciseChunking method shown in Table 1 of our paper), such "sudden transfer" can affect the results. However, after calibrating the uncertainty using SNR, we observed significant performance improvements. Furthermore, our experiments demonstrate that even under the influence of such "sudden transfer," our method can still outperform previous approaches with significantly lower training costs.
>
>
>
>
> **Question 2**: What is the role of N as defined in line 285?
>
>
> **Response 2**: Thank you very much for your reminder! The use of N here is unnecessary and was a typo. In fact, this step involves using BM25 to coarsely filter and select M semantically relevant chunks for each chunk, aiming to reduce the computational cost of subsequent measurements with the LLM. Thank you again! We have corrected this typo in the paper.
>
>
> **Question 3**: How did you select the training and test sets from the LongBench suite?
>
>
> **Response 3**: Thank you for the insightful question. In fact, we carefully considered the diversity of both the training and test datasets when making our selections. Specifically:
>
>
> 2WikiMultihopQA and Musique are tasks that require answering questions based on multiple provided documents.
>
>
> TREC is a classification task that involves categorizing questions into 50 predefined categories.
>
>
> TriviaQA is a single-document question-answering task with several few-shot examples provided.
>
>
> SAMSum is a dialogue summarization task, also including few-shot examples.
>
>
> HotpotQA involves answering questions based on multiple given documents.
>
>
> MultiFieldQA focuses on answering questions in English based on a single document, with documents spanning diverse fields.
>
>
> Qasper is centered around questions posed on a single research paper, where the questions are generated by NLP readers and answered by NLP practitioners.
>
>
> NarrativeQA involves answering questions based on stories or scripts, requiring an understanding of key elements such as characters, plots, and themes.
>
>
> QMSum is a summarization task that requires summarizing meeting transcripts based on a user query.
>
>
> **To further address your concerns about the sensitivity of dataset selection, we have added experiments in our response to Question 4, where we trained and tested the model using newly selected training and test datasets.**

---

> ### Author Response · Authors · 2024-11-24
> **Response to Reviewer Q3d1 (Part 2/3)**
>
> **Question 4**: Not all training and test sets feature distributional shifts in their contexts. For example, both HotpotQA and 2WikiMQA use Wikipedia as their source of context.
>
>
> **Response 4**: Thank you very much for your reminder! We have reselected the training and test datasets. In the new experiments, the training set includes MultiFieldQA, Musique, QMSum, TREC, and TriviaQA, while the test set consists of NarrativeQA, Qasper, SAMSum, and 2WikiMultihopQA. The detailed experimental results are presented below.
> | LLaMA-2-7B-Chat-HF   | Truncate |BERT	 | Contriever| BGE-M3| BGE-Large| GRAGON-PLUS|Ours|
> |  ----  | :----:  | :----:  | :----:  | :----:  | :----:  | :----:  | :----:  |
> | 2WikiMultihopQA|28.50		| 32.73  |33.60		 |29.64  |34.14     |34.61       |**37.31**
> | NarrativeQA    |17.31		| 17.58  |16.91		 |19.45  |19.46     |20.01       |**20.20**
> | Qasper 		 |18.14		| 19.07  |**21.01**       |20.31  |19.93     |20.50       |20.81
> | SAMSum         |40.45		| 40.01  |38.45		 |40.37  |40.34     |41.08       |**41.26**
> | Average        |26.10		| 27.34  |27.49		 |27.44  |28.47     |29.05       |**29.90**
> **Table 1**: The results of new experiments
>
> All experiments were conducted on LLaMA-2-7B-Chat-HF. As observed, our method continues to achieve the best performance on the new datasets. This demonstrates the robustness of our method, as it does not rely on the specific selection of datasets.
>
>
>
> **Question 5**: In line 451, how do you conclude that "RSA is an effective metric for assessing distribution shifts," given that you only have two data points (GSM8K and 2WikiMQA)?
>
>
> **Response 5**: Thank you very much for your reminder. In fact, Figure 3 in the original paper includes three results, representing the original training set, GSM8K, and 2WikiMQA. Additionally, we have obtained results on other test sets as well, but we opted to include only one in the figure for aesthetic reasons. However, we did not consider that this might raise concerns for readers. Once again, thank you for pointing this out. Here, we will present the RSA results on two additional datasets in the anonymous link below: https://ufile.io/uvmt2qsi
>
>
> **Question 6**: The "10" mentioned in line 310 should be k
>
>
> **Response 6**: Thank you for your suggestion. We have made the corresponding changes in the paper.
>
>
>
>
> **Weakness:**
>
> **Weakness 1**: Some content is difficult to understand. For example, in line 245, "The probability distribution within this window is stable," which does not seem to imply a stable distribution. Additionally, in Section 4.1, the phrase "number of chunks" is unclear, as the previous mention of the same term refers to the total number of chunks in the training set.
>
>
> **Response to weakness 1**:
>
> (1) Here, I believe directly presenting a specific example might be more helpful for understanding. Below is an anonymous code link: https://ufile.io/56d9jxyh
>
> (2) Thank you very much for your reminder. The "number of chunks" here refers to the quantity of chunks retrieved during the retrieval process. Based on your suggestion, we have revised line 420 in the original text from "Ablation Study on Chunk Length and The Number of Chunks" to "Ablation Study on the Length and Number of Retrieved Chunks."
>
>
>
>
> **Weakness 2**: The definition of the negative samples is questionable. It appears that the negative samples are among the top M most similar chunks to the anchor. If this is the method for selecting hard negative examples, the hyperparameters in the paper would play a crucial role.
>
>
> **Response to weakness 2**: Here, we followed the settings from previous works, such as EPR[1] and CEIL[2], without making any modifications to these hyperparameters.

---

> ### Author Response · Authors · 2024-11-24
> **Response to Reviewer Q3d1 (Part 3/3)**
>
> **Weakness 3**: Numerous hyperparameters are introduced, such as the threshold σ, the number of samples M and m, the chunk size, and the sliding window size. However, the authors do not explain how these hyperparameters were chosen or how sensitive the model is to these values.
>
>
> **Response to weakness 3**: Thank you for the question. In the anonymous code link: https://ufile.io/56d9jxyh, it can be observed that the choice of threshold is not particularly sensitive. When adjusting the sliding window size, one only needs to observe the data accordingly and modify the threshold selection. Experiments with different chunk sizes are presented in Table 3 of our original paper. All other parameters were adopted from previous works[1][2] without modification.
>
>
>
> **Weakness 4**: It is challenging to conclude that the method can be integrated with any LLMs, given that uncertainty is only calculated using Llama 2 7B. The tested models also use Llama 2 as their base, which shares the same pretraining data. From Table 1, it is apparent that when the uncertainty score model does not match the LLM, the performance gain from the proposed method decreases. This raises doubts about whether the retriever must match the LLM used.
>
>
> **Response to weakness 4**:
> (1) Thank you for the insightful question. Firstly, since our measuring of the uncertainty inherently reflects the model's perplexity, it is naturally most effective when the retriever and LLM are well-matched. However, this does not mean that our method cannot be integrated with other LLMs. To demonstrate this, we conducted additional experiments using Vicuna-7B to measure uncertainty and performed RAG experiments on LLaMA-2-7B-Chat-HF. The results are presented in the table below. As observed, although the improvements are not as significant as those achieved when using LLaMA-2-7B-Chat-HF itself for measuring the uncertainty (which is expected), the performance still surpasses other baselines.
>
>
> | LLaMA-2-7B-Chat-HF 	     | Truncate |BERT	 | Contriever| BGE-M3| BGE-Large| GRAGON-PLUS|Ours(Vicuna-7B)|
> |  ----  | :----:  | :----:  | :----:  | :----:  | :----:  | :----:  | :----:  |
> | 2WikiMultihopQA|28.50		|32.73   |33.60      |29.64  |34.14     |34.61       |**35.74**
> | Musique    	 | 9.41		|18.74   |14.25      |**24.27**  |24.20     |20.50       |22.68
> | TREC			 |64.50		|66.00   |70.00      |**71.00**  |70.50     |70.50       |67.50
> | Triviaqa		 |77.80		|78.69   |76.09      |75.74  |75.10     |77.51       |**79.46**
> | SAMSum		 |40.45		|40.01   |38.45      |40.37  |40.34     |41.08       |**42.70**
> | Average		 |44.13		|47.23   |46.48      |48.20  |48.85     |48.84       |**49.62**
> **Table 2**: The results of using Vicuna-7B as the uncertainty measurement model and testing on LLaMA-2-7B-Chat-HF.
>
>
> (2) **More importantly**, we would like to point out that the statement "it is apparent that when the uncertainty score model does not match the LLM, the performance gain from the proposed method decreases" **is not entirely rigorous**. As demonstrated, even when **comparing two baselines** (e.g., Contriever and GRAGON-PLUS), the performance differences vary across different LLMs. Specifically, on LLaMA-2-7B-Chat-HF, GRAGON-PLUS outperforms Contriever by an average of **2.36%**; on LLaMA-2-13B-Chat-HF, this advantage decreases to **0.78%**; and on Vicuna-7B, it further drops to **0.39%**！ This indicates that the improvements achieved by retrieval models inherently differ across various LLMs.
>
> **If there are any questions, please let us know. And if you think that we have addressed your concerns, could you please consider raising the score? Thank you very much for your support.**
>
>
> [1] Ohad Rubin, Jonathan Herzig, and Jonathan Berant. Learning to retrieve prompts for in-context learning. arXiv preprint arXiv:2112.08633, 2021.
>
> [2] Jiacheng Ye, Zhiyong Wu, Jiangtao Feng, Tao Yu, and Lingpeng Kong. Compositional exemplars for in-context learning. arXiv preprint arXiv:2302.05698, 2023.

---

> ### Comment · Reviewer_Q3d1 · 2024-11-25
>
> Thank you for your detailed response. It addresses some of my initial questions and provides clarity on certain aspects of the paper. However, there are still some remaining issues that I would like to highlight:
>
> 1. Integration of New Results and Explanations:
> For the points raised that could significantly impact the soundness of the paper—such as the details on the chunking procedure, the hyperparameter selection process, sensitivity analysis, experiments with alternative training/test set combinations, and the additional results in Figure 3—I would like to understand how these new results and explanations will be incorporated into the paper. It would enhance the paper’s clarity and utility for readers if these details were included directly in the main text or appendices rather than being confined to the accompanying code. Would you consider providing a revised version of the paper reflecting these updates?
>
> 2. Clarification on Q3 - Dataset Selection:
> Regarding my earlier question on the choice of training/test set combinations, I was seeking an explanation of the rationale behind selecting these specific combinations, rather than a description of the datasets themselves. Reviewer VFfy also raised a similar concern (Q5). Are these combinations based on specific criteria, and have you considered experimenting with a more diverse or simple mix of datasets to assess the generality of your results?
>
> 3. Further Questions on Q5 - Metric Effectiveness:
> * My original concern was that the conclusion, "$x$ is an effective metric for $y$," is based on only three results without any additional quantitative analysis. This makes it challenging to substantiate the claim. Moreover, using RSA as a metric to demonstrate that the training set exhibits the most "distributional shift" among the three datasets seems to introduce a degree of circular reasoning. Could you clarify or address this concern?
> * Additionally, in the new results provided, the curve for TriviaQA appears to fall below that of the training set. Do you have an explanation for this behavior, and how does it impact your conclusions?
>
> I hope this feedback is helpful and contributes to strengthening the paper’s clarity and rigor. Thank you again for your efforts in responding to my earlier questions.

---

> > ### Author Response · Authors · 2024-11-29
> > **Response to Reviewer Q3d1 (Part 1/2)**
> >
> > Dear Reviewer Q3d1,
> >
> > Thank you very much for your comment, we will address your concerns below.
> >
> >
> > **Question 1:** Thank you for your detailed response. It addresses some of my initial questions and provides clarity on certain aspects of the paper. However, there are still some remaining issues that I would like to highlight:
> > Integration of New Results and Explanations: For the points raised that could significantly impact the soundness of the paper—such as the details on the chunking procedure, the hyperparameter selection process, sensitivity analysis, experiments with alternative training/test set combinations, and the additional results in Figure 3—I would like to understand how these new results and explanations will be incorporated into the paper. It would enhance the paper’s clarity and utility for readers if these details were included directly in the main text or appendices rather than being confined to the accompanying code. Would you consider providing a revised version of the paper reflecting these updates?
> >
> >
> > **Response to question 1:** Thank you very much for your suggestions! We have added the new experimental results and some additional analyses to the appendix. In the main text, we have pointed readers to the specific appendix sections for areas that may raise questions (such as the explanation of stability and the analysis of RSA). Additionally, we have made revisions to the manuscript based on the writing suggestions provided by the reviewers. The latest version of our paper has been uploaded to an anonymous link: https://ufile.io/9apje264. To facilitate review, the modified and added sections have been highlighted in blue.
> >
> > **Question 2:** Clarification on Q3 - Dataset Selection: Regarding my earlier question on the choice of training/test set combinations, I was seeking an explanation of the rationale behind selecting these specific combinations, rather than a description of the datasets themselves. Reviewer VFfy also raised a similar concern (Q5). Are these combinations based on specific criteria, and have you considered experimenting with a more diverse or simple mix of datasets to assess the generality of your results?
> >
> > **Response to question 2**: Thank you for the insightful question. In fact, we initially did not spend much time selecting the datasets. We believed that as long as the datasets met basic diversity criteria and there was not a significant disparity between the training and test sets, they would be acceptable. At the same time, we would like to emphasize that **the dataset choices for our new experiments were made entirely based on the reviewers' feedback**. For example, your Question 4 ("Not all training and test sets feature distributional shifts in their contexts. For example, both HotpotQA and 2WikiMQA use Wikipedia as their source of context.") led us to remove HotpotQA from the training set. Additionally, Reviewer jHwd's Weakness 2 ("The evaluation tasks should include single-doc tasks such as NQ and Qasper. To still keep the distribution shifts, you can move these two tasks from the training set to the evaluation set, and add the three few-shot learning tasks into the training data.") prompted us to move NQ and Qasper from the training set to the test set, while adding the few-shot learning tasks to the training set. This entire process did not involve any special or additional selection on our part, so we can confidently state that our approach remains robust with respect to the choice of training and test sets. Alternatively, if you have any further suggestions regarding the construction of this dataset, we would be very happy to discuss them with you!

---

> > ### Author Response · Authors · 2024-11-29
> > **Response to Reviewer Q3d1 (Part 2/2)**
> >
> > **Question 3:** Further Questions on Q5 - Metric Effectiveness:
> >
> > (1) My original concern was that the conclusion, "x is an effective metric for y," is based on only three results without any additional quantitative analysis. This makes it challenging to substantiate the claim. Moreover, using RSA as a metric to demonstrate that the training set exhibits the most "distributional shift" among the three datasets seems to introduce a degree of circular reasoning. Could you clarify or address this concern?
> > (2) Additionally, in the new results provided, the curve for TriviaQA appears to fall below that of the training set. Do you have an explanation for this behavior, and how does it impact your conclusions?
> >
> > **Response to question 3:**
> > (1) Thank you for your question. First, let us clarify two concepts: **"distribution shift of the dataset"** and **"shift in the representation space"**. The former refers to the difference in the distribution of the dataset itself, which is a constant and inherent shift. The latter refers to the difference between the distribution of the dataset in the representation space after the model has been trained, compared to the distribution before training.
> >
> > In fact, our aim is not to “demonstrate that the training set exhibits the most "distributional shift" among the three datasets.”“ What we presented in Figure 3 are three datasets with intuitively decreasing distributional shifts: GSM8K, a dataset with a significant "distribution shift" (GSM8K is a math problem dataset, and we believe its distribution shift is the largest, which does not require additional proof), followed by 2WikiMultihopQA (which exhibits a smaller distribution shift than GSM8K), and finally the original training set (which has no distribution shift). We clearly observe that as the distribution shift of the dataset decreases, the shift in the representation space of our model increases (as reflected in the figure, where the gap between the pre-trained and post-trained models grows larger. This is also easy to explain, as the goal of our training is to obtain a better representation space. Therefore, when the distribution is closer to the training set, our model naturally exhibits a larger shift in the representation space.). Therefore, we argue that RSA, which captures shifts in the representation space, can be used to assess distribution shifts.
> > To further substantiate our hypothesis, we performed an additional analysis on the MATH dataset, which intuitively has a larger distributional shift, as can be seen in https://ufile.io/9apje264 Appendix G
> >
> > (2) Thank you very much for your reminder! This was an oversight on our part. Our original intention was to show you the dataset MultiFieldQA, which is included in the training set, and the dataset Musique, which is not included in the training set, in order to further validate our conclusions. However, due to this oversight, we mistakenly labeled MultiFieldQA as TriviaQA. We have now provided the correct results for these three datasets (MultiFieldQA, TriviaQA, and Musique) in this anonymous link: https://ufile.io/zktmefl6 . Thank you again for your helpful reminder!
> >
> > **Once again, thank you for all of your suggestions! If you think that we have addressed your concerns, could you please consider raising the score? Thank you very much for your support!**

---

> > ### Author Response · Authors · 2024-12-02
> > **Waiting for further discussion**
> >
> > Dear Reviewer Q3d1,
> >
> > Thank you very much for your invaluable feedback on our paper. We have meticulously reviewed each of your points and endeavored to address them thoroughly. We would greatly appreciate it if you could review our responses and let us know if you have any additional comments regarding the paper or the rebuttal. We are eager to embrace all your critiques and integrate them into our work.
> >
> > If you think that we have addressed your concerns, could you please consider raising the score? Thank you very much for your support.
> >
> > Best regards,
> >
> > The Authors

---

> ### Author Response · Authors · 2024-12-03
>
> Dear Reviewer Q3d1,
>
> In our most recent response, we addressed the concerns you raised previously. However, it is possible that for various reasons, our response may not have reached you. If you find that our previous answers have resolved your concerns, would you kindly consider raising your score? We greatly appreciate your support.
>
> Thank you very much.
>
> Best regards,
>
> The Authors

---

### Meta-Review · Area_Chair_1m29 · 2024-12-24

**Metareview:**

The paper introduces a novel method for estimating similarity between text chunks in retrieval-augmented generation (RAG). This method is based on what the authors refer to as a “signal-to-noise ratio” (SNR) measure, which is computed after feeding a large language model (LLM) with two chunks and analyzing the output probabilities. This serves for computing  a similarity metric between the chunks and then construct a dataset of positive and negative examples for training a retrieval model with contrastive loss. During inference, an input document is divided into chunks, and the model retrieves the most similar chunks for a given query. These retrieved chunks, along with the query, are then input into the LLM to generate the final answer. To enhance the model’s generalization under distribution shifts, the authors proposed two data scaling strategies. Experiments on various datasets demonstrate state-of-the-art (SOTA) performance while using less training data compared to traditional RAG approaches.

The reviewers acknowledged the novelty of the idea and the strong performance of the proposed model. However, they raised concerns about the lack of clear motivation for the SNR approach, insufficient clarity in the technical description, and limited experimental comparisons. In response, the authors provided extensive clarifications and additional experimental results, significantly improving the initial version. Nevertheless, the substantial revisions needed to incorporate these changes mean that the paper would benefit from further refinement in terms of organization and writing and is not yet ready for acceptation.

**Additional Comments On Reviewer Discussion:**

The authors provided extensive new experimental results during the rebuttal, which strengthened the initial submission. However, this did not change the reviewers' opinion regarding acceptance.

---

### Decision · Program_Chairs · 2025-01-22

Reject